# A Watermark for Low-entropy and Unbiased Generation in Large Language Models

## Abstract

Recent advancements in large language models (LLMs) have highlighted the risk of misusing them, raising the need for accurate detection of LLM-generated content. In response, a viable solution is to inject imperceptible identifiers into LLMs, known as watermarks. Previous work demonstrates that unbiased watermarks ensure unforgeability and preserve text quality by maintaining the expectation of the LLM output probability distribution. However, previous unbiased watermarking methods suffer from one or more of the following issues: (1) requiring access to white-box LLMs during detection, (2) incurring long detection time, (3) being not robust against simple watermarking attacks, (4) failing to provide statistical guarantees for the type II error of watermark detection, and (5) being not statistically unbiased for low-entropy scenarios, which hinder their deployment in practice. This study proposes the Sampling One Then Accepting (STA-1) method, a watermark that can address all of these issues. Moreover, we discuss the tradeoff between watermark strength and text quality for unbiased watermarks. We show that in low-entropy scenarios, unbiased watermarks face a tradeoff between watermark strength and the risk of unsatisfactory outputs. Experimental results on both low-entropy and high-entropy datasets demonstrate that STA-1 achieves text quality and watermark strength comparable to existing unbiased watermarks, with a low risk of unsatisfactory outputs. Implementation codes for this study are available online (hidden for peer review).

## 1 Introduction

Large language models (LLMs) are large-scale deep learning models that can understand and generate natural languages by learning from a large amount of textual data. Typical generative LLMs, such as ChatGPT (Ouyang et al., 2022) and LLaMA (Touvron et al., 2023), can answer questions, translate languages, and create codes with qualities comparable to humans. As LLMs can generate contents more efficiently at a lower cost compared to humans, the risk of LLMs being employed to generate biased, fake, or malicious contents is also increasing (Mirsky et al., 2023). For example, LLMs may exhibit biased information against underrepresented groups of people (Abid et al., 2021; Fang et al., 2024), create misinformation (Pan et al., 2023), and harm academic integrity (Zhao et al., 2023). To reduce the harm caused by LLMs, identifying LLM-generated content precisely and efficiently becomes a crucial issue (Kirchenbauer et al., 2023a).

Watermarks are identifiers imperceptible to humans but detectable by certain models (Liu et al., 2023b). In the era of LLMs, a watermarking method is a strategy to control the randomness of token generation by LLMs (Kirchenbauer et al., 2023a; Fernandez et al., 2023), with the randomness preserved confidentially by LLM owners. In practice, watermarks should have unforgeability against deciphering watermarking generation attacks (Liu et al., 2023b). A watermarking method demonstrates the ability against forgeries if it can hide the distinguishability between the original unwatermarked text and its watermarked counterpart (Christ et al., 2023; Liu et al., 2023b). Thus, it is required that a watermarking method adjusts the probability distribution while maintaining the same expectation as the unwatermarked distribution (Hu et al., 2024; Kuditipudi et al., 2023), defined as ***unbiased*** watermarks.

Existing unbiased watermarks can be categorized according to the stage where watermarks are injected: distribution reweighting and controlled sampling (Liu et al., 2023b). For distribution

reweighting, Hu et al. (2024) proposes $\gamma$-reweight, which uses the log-likelihood ratio (LLR) test by comparing the likelihood of the text produced by watermarked and unwatermarked white-box LLMs. It requires the prompt as input and a white-box LLM in watermark detection (Fernandez et al., 2023; Hu et al., 2024). Also, the watermark is unstable because changing the first token of the generated text can lead to huge deviations from the original likelihood value (Fernandez et al., 2023). In response, Wu et al. (2024) avoid the LLR test and propose Dipmark, an extension of $\gamma$-reweight with more general parameter settings. However, although both $\gamma$-reweight and Dipmark ensure the type I error of watermark detection, they fail to provide statistical guarantees for the type II error (Hu et al., 2024; Wu et al., 2024). For controlled sampling, Christ et al. (2023) introduce a watermarking method that uses a sequence of random values to guide the token sampling process. However, their method is not robust enough against simple removal attacks (Liu et al., 2023b). Kuditipudi et al. (2023) also use random values to control the sampling and introduce a permutation test on detection that does not require white-box access to LLMs. However, this permutation test is time-consuming theoretically and empirically. Fairoze et al. (2023) propose to sample the token sequence generation until its hash matches a key value. According to their distortion-free definition, the upper bound of the difference between probabilities before and after watermarking is $\exp(-a)$, where $a$ is the minimal entropy. The difference is not negligible in low-entropy scenarios. Note that using random values to control sampling can be treated as a special case of distribution reweighting where only the probability of the sampled token is reweighted to 1 (Kuditipudi et al., 2023). Thus, we build our analysis framework in Section 2 solely based on distribution reweighting.

We conclude the research gaps in Table 5 in Appendix A. In response to the challenges, we propose the Sampling One Then Accepting (STA-1) method that can simultaneously overcome the above gaps. STA-1 traces back to the original watermarking method (denoted as KGW) (Kirchenbauer et al., 2023a) where the token set is divided into a green and a red list at each generation step. Instead of raising logits in the green list, STA-1 samples a token from the original probability distribution and accepts it if it is in the green list. If the sampled token is in the red list, it resamples another token and accept it. By counting the number of green list tokens, it employs the $z$-test for watermark detection, which naturally addresses the issues encountered with the LLR test and eliminates the need for prompts and white-box LLMs in detection. Simultaneously, the detection of STA-1 is efficient and only requires $O(m)$ time complexity, where $m$ is the number of tokens. The STA-1 method is also robust against simple insertion and removal attacks because changing tokens can only affect the detection score around these tokens. Meanwhile, we prove that STA-1 is unbiased and provides statistical guarantees for the type II error. More interestingly, the bounds are linked to the Gini index of the probability distribution, which is a common metric in machine learning (Breiman, 2017) compared to the proposed Spike entropy in previous work (Kirchenbauer et al., 2023a).

In this study, we also clarify the watermark strength and text quality tradeoff in unbiased watermarks. The KGW method faces a tradeoff between watermark strength and text quality, which means a higher detection power results in a lower text quality (Kirchenbauer et al., 2023a). Previous work claims that unbiased watermarks can avoid this tradeoff given the preserved text quality by maintaining the expectation of probability distribution (Hu et al., 2024). We challenge this claim by considering a simple low-entropy scenario, where we show that unbiased watermarks still face a tradeoff between watermark strength and text quality. However, under the same expectation constraint, the text quality is related to the risk of unsatisfactory outputs. Specifically, unsatisfactory outputs in low-entropy scenarios represent that the watermarking method alters probability distribution too much such that high-probability tokens cannot be sampled at risk. We discuss the risk via the variance of the probability after altering, which is a common practice of risk-return analysis (Sharpe, 1998). We prove that STA-1 is less risky than previous unbiased watermarks. Moreover, we propose STA-M, an extension of STA-1, by setting up a threshold for entropy in generation (Lee et al., 2023; Wang et al., 2023) and sampling more times for high-entropy steps. Although STA-M is not unbiased theoretically, it allows higher watermark strength with small performance shifts empirically. Also, STA-M is robust against various watermarking attacks. We summarize our contributions as follows:

1. We propose STA-1, an unbiased watermarking method that is practical and has statistical guarantees on type II error of watermark detection. Moreover, we introduce STA-M, an extension of STA-1 that enhances watermark strength with low text quality shifts.

2. We clarify the watermark strength and text quality tradeoff in unbiased watermarks. In low-entropy scenarios, the text quality is related to the risk of unsatisfactory outputs. We show that STA-1 has a lower risk theoretically compared to other unbiased watermarks.

3. Experimental results on public low-entropy and high-entropy datasets empirically show that STA-1 achieves comparable performances against other unbiased watermarks and has a low risk of unsatisfactory outputs. Meanwhile, STA-M demonstrates high watermark strength in the low-entropy dataset and robustness against different watermarking attacks.

## 2 PRELIMINARY

**Notations.** We follow notations in previous work (Kirchenbauer et al., 2023a; Hu et al., 2024) to represent the generation task of LLMs. Let $P_M$ denote a pretrained LLM and $\mathcal{V}$ is the overall token (vocabulary) set. An example token set contains more than 50,000 tokens ($|\mathcal{V}| > 50000$) (Radford et al., 2019). For simplicity, we use Python-style notation for an ordered token sequence, where $x^{-m:n} = (x^{-m}, x^{-m+1}, \cdots, x^n)$, $m$ and $n$ are integers. In a typical LLM generation task, an LLM receives a sequence of $N_p + 1$ tokens $x^{-N_p:0}$, known as a prompt, and outputs a sequence of $T$ tokens $x^{1:T}$ step by step. At step $t$, the probability of each token in the token set $\mathcal{V}$ is given by the conditional distribution $P_M(x^t|x^{-N_p:(t-1)})$. The LLM generation follows an autoregressive fashion, where the joint probability of the generated tokens are as $P_M(x^{1:T}|x^{-N_p:0}) = \prod_{t=1}^{T} P_M(x^t|x^{-N_p:(t-1)})$.

When applying watermarking techniques, the LLM employs a private key $k$ to adjust the conditional distribution from $P_M(x^t|x^{-N_p:(t-1)})$ to $P_{M,w}(x^t|x^{-N_p:(t-1)};k)$, where $P_{M,w}$ indicates a watermarked model and the private key $k$ is randomly selected from a key space $K$ according to a known distribution $P_K(k)$. An unbiased watermark requires that the expectation of the watermarked distribution equals that of the original distribution (Hu et al., 2024), defined as follows.

**Definition 1** (Unbiased watermark). Given a prompt $x^{-N_p:0}$ and a known distribution $P_K(k)$ of the key $k$, a watermarking method is unbiased towards the original model $P_M$ if the watermarked model $P_{M,w}$ satisfies

$$\mathbb{E}_{k \sim P_K(k)} \left[ P_{M,w}(x^t|x^{-N_p:(t-1)};k) \right] = P_M(x^t|x^{-N_p:(t-1)}), \tag{1}$$

for any prompt $x^{-N_p:0} \in \mathcal{V}^{N_p+1}$, any token $x^t \in \mathcal{V}$, and all generation steps $1 \leq t \leq T$.

**Previous distribution reweighting methods.** Since controlled sampling can be viewed as a special case of distribution reweighting, we build our analysis framework based on distribution reweighting. Formally, a reweighting function $R_k : \mathcal{P}_\mathcal{V} \to \mathcal{P}_\mathcal{V}$ maps from $P_M(x^t|x^{-N_p:(t-1)})$ to $P_{M,w}(x^t|x^{-N_p:(t-1)};k)$, where $\mathcal{P}_\mathcal{V}$ denotes the probability distribution space over the vocabulary set $\mathcal{V}$. A reweighting method $R : K \times \mathcal{P}_\mathcal{V} \to \mathcal{P}_\mathcal{V}$ contains all realized reweighting functions $R_k$ among the key space $k \in K$. Following Definition 1, $R$ is an unbiased reweighting method if $\mathbb{E}_{k \sim P_K(k)} [R_k(P_M)] = P_M$. Next, we introduce previous watermarking methods (Kirchenbauer et al., 2023a; Hu et al., 2024; Wu et al., 2024; Kuditipudi et al., 2023) and refer readers to Appendix B for more details.

**KGW (Kirchenbauer et al., 2023a):** The KGW method (Kirchenbauer et al., 2023a) randomly splits the vocabulary set $\mathcal{V}$ into a green list and a red list based on a uniformly distributed key $k$. The soft KGW method adds a predefined constant $\delta$ to the green list tokens' logits while keeping the red list tokens' logits fixed.

**Dipmark (Wu et al., 2024) and $\gamma$-reweight (Hu et al., 2024):** Wu et al. (2024) propose an unbiased watermarking method named Dipmark. Dipmark shuffles all probability masses $P_M(x^t|x^{-N_p:(t-1)})$ over the vocabulary set within the probability interval $[0, 1]$ based on a key $k$. A hyperparameter $\alpha \in [0, 0.5]$ partitions the interval $[0, 1]$ into three segments: $[0, \alpha]$, $(\alpha, 1 - \alpha]$, and $(1 - \alpha, 1]$. Probabilities in the first segment are set to 0, those in the second remain constant, and those in the third are doubled. Dipmark becomes $\gamma$-reweight when $\alpha = 0.5$.

**RDW (Kuditipudi et al., 2023):** We focus on the RDW method via an inverse transform sampling scheme. Given a uniformly distributed key $k$, RDW first shuffles all probability masses within the interval $[0, 1]$, then it randomly samples a value $u \sim U(0, 1)$. Here, $u$ is viewed as the cumulative distribution function value of $P_M(x^t|x^{-N_p:(t-1)})$ with respect to the permutation; it is subsequently inverse transformed to generate a token. The probability of the sampled token is reweighted to 1, while the probabilities of all other tokens are reweighted to 0.

## 3    A SIMPLE PROTOCOL FOR A LOW-ENTROPY SCENARIO

The low-entropy text refers to a relatively deterministic sequence in natural language. The entropy measures the uncertainty of the probability distribution $P_M(x^t|x^{-N_p:(t-1)})$ at a single generation step among the token set $\mathcal{V}$, where low entropy means low uncertainty. For example, in code writing, the structure of a code sequence is regularized where few changes can be made (Lee et al., 2023). More explicitly, for a typical English pangram such as 'The quick brown fox jumps over the lazy dog' (Kirchenbauer et al., 2023a), both humans and machines should generate similar if not identical output. For example, when provided with the prompt 'The quick brown fox jumps over the lazy', the trained LLaMA-2-7B (Touvron et al., 2023) outputs an empirical probability above 0.8 for the next word 'dog'.

**Problem modeling.** Low-entropy scenarios exist in text generation tasks of LLMs. We aim to model a simple problem protocol for the low-entropy generation scenario. For simplicity, we consider the low-entropy scenario where only one token probability is significantly large. Specifically, denote $p_{max}$ as the largest probability of a token in the probability distribution $P_M(\cdot|x^{-N_p:(t-1)})$. We make an intuitive assumption that except $p_{max}$, other $|\mathcal{V}| - 1$ probabilities are small enough to uniformly fill in the remaining $1 - p_{max}$ probability value.

## 4    METHOD: SAMPLING THEN ACCEPTING

In this section, we propose the Sampling One Then Accepting (STA-1) method, and discuss detecting the STA-1 generated text using the $z$-test. Theoretically, we show that STA-1 is unbiased and its type II error of the $z$-test has statistical guarantees. Next, we analyze previous unbiased watermarks and STA-1 under the low-entropy protocol in Section 3. We finally introduce Sampling M Then Accepting (STA-M), an extension of STA-1.

### 4.1    SAMPLING ONE THEN ACCEPTING

We start by proposing the Sampling One Then Accepting (STA-1) method in Algorithm 1, which is always unbiased and easy to analyze. First, the hash value of the last generated token is computed and employed as the seed of a random number generator (RNG). We use the RNG to divide the token set into a green and a red list (Kirchenbauer et al., 2023a). Next, we sample from the original LLM output distribution (as depicted in Line 4 of Algorithm 1), accept the sampling if the token is in the green list (as depicted in Line 5 and Line 6 in Algorithm 1), sample again if the token is in the red list (as depicted in Line 7 and Line 8 in Algorithm 1), and the second sampling is always accepted.

---

**Algorithm 1** STA-1 Text Generation

**Input:** A pretrained LLM $P_M$, a watermark key $k \in K$, the proportion of the green list $\gamma \in (0, 1)$, and a prompt $x^{-N_p:0}$

1: **for** $t = 1, 2 \ldots, T$ **do**
2:    Get the probability distribution of tokens $p^t = P_M(\cdot|x^{-N_p:(t-1)})$
3:    Compute the hash of the last token $x^{t-1}$. Partition the token set $\mathcal{V}$ to form the green $G$ and red $R$ list based on key $k$, the hash, and the proportion $\gamma$
4:    Sample the candidate token $x_c^t$ with $p^t$
5:    **if** $x_c^t \in G$ **then**
6:       Accept the sampling, the next generated token $x^t = x_c^t$
7:    **else**
8:       Deny the sampling, sample $x^t$ from the distribution $p^t$
9:    **end if**
10: **end for**

**Output:** The generated text $x^{1:T}$

---

STA-1 is a simple but effective watermarking method. The properties of STA-1 include: (1) STA-1 is an unbiased watermark; (2) The number of green list tokens in STA-1 generated texts has a lower bound on its mean and an upper bound on its variance, which further provides explicit statistical guarantees for the type II error in the STA-1 detection test; (3) STA-1 has a lower risk for low-entropy

generation compared to previous work. In deriving theoretical results, we assume that the key $k$ is randomly sampled from a uniform distribution. Therefore, the random partition of green and red lists associated with this key is also uniform (Kirchenbauer et al., 2023a). We start by analyzing the unbiased characteristic of STA-1.

**Theorem 1.** *The STA-1 method (Algorithm 1) is an unbiased watermark.*

*Proof. See Appendix C.1.*

### 4.1.1 STATISTICAL TEST GUARANTEES OF STA-1

**Detecting the STA-1 generated text.** The detection of STA-1 compares the empirical proportion of green list tokens in the given text against the green list proportion $\gamma$ (Kirchenbauer et al., 2023a). We employ the $z$-test where the null hypothesis ($H_0$) is that the text is generated without knowing the green-red list partition. Denote $|S|_G$ as the number of green list tokens in this text. Under $H_0$, $|S|_G$ follows a Bernoulli distribution $B(T, \gamma)$ with a mean of $\gamma T$ and a variance of $\gamma(1 - \gamma)T$. The $z$-score is calculated with the empirical $|S|_G$ as

$$z = \frac{|S|_G - \gamma T}{\sqrt{\gamma(1-\gamma)T}}. \tag{2}$$

The alternative hypothesis ($H_a$) is that the text is generated with STA-1. Under $H_a$, $|S|_G$ is expected to be larger than $\gamma T$. We can detect watermarked texts with a certain confidence level if the $z$-score exceeds a $z$ threshold. For example, if $z > 2$, we are more than 97.7% confident that the text is watermarked under the one-tail test.

To ensure the effectiveness of the $z$-test, under $H_a$, a lower bound on the expectation of $|S|_G$ and an upper bound on the variance of $|S|_G$ are required. We establish the necessary lower and upper bounds in the following theorem. Because both bounds are related to the Gini index of the LLM output distribution, we define the Gini index first.

**Definition 2** (Gini index). Given a discrete probability distribution $p = (p_1, p_2, \cdots, p_N)$, the Gini index of $p$ is defined as

$$Gini(p) = \sum_{i=1}^{N} p_i(1 - p_i). \tag{3}$$

A low Gini index implies less uncertainty in the probability distribution, resulting in a low-entropy scenario. Next, we propose the mean and variance bounds of $|S|_G$.

**Theorem 2.** *For STA-1 generated text sequences with $T$ tokens, let the random green list have a fixed size of $\gamma|\mathcal{V}|$, and $p_i^t$ denote the LLM's raw output probability of the $i$-th token in $\mathcal{V}$ at step $t$, $i = 1, 2, \cdots, |\mathcal{V}|$, $p^t = (p_1^t, p_2^t, \cdots, p_{|\mathcal{V}|}^t)$. If an STA-1 generated sequence $S$ has an average Gini index larger than or equal to $Gini^*$, that is,*

$$\frac{1}{T}\sum_{t=1}^{T} Gini(p^t) = \frac{1}{T}\sum_{t=1}^{T}\sum_{i=1}^{|\mathcal{V}|} p_i^t(1 - p_i^t) \geq Gini^*.$$

*Then the expectation of $|S|_G$ is at least*

$$\mathbb{E}(|S|_G) \geq \gamma T + (1 - \gamma)\gamma T Gini^*. \tag{4}$$

*With one additional assumption that $\gamma$ and $Gini^*$ satisfy $\gamma + (1 - \gamma)\gamma Gini^* \geq 0.5$, the variance of $|S|_G$ is at most*

$$\mathbb{V}(|S|_G) \leq T[\gamma + (1 - \gamma)\gamma Gini^*][1 - \gamma - (1 - \gamma)\gamma Gini^*]. \tag{5}$$

*Proof. See Appendix C.2.*

**Remark 1.** The additional assumption required for the variance upper bound, $\gamma + (1 - \gamma)\gamma Gini^* \geq 0.5$, implies that a larger green list is necessary in low-entropy scenarios to establish an upper bound on the variance of $|S|_G$. By selecting $\gamma \geq 0.5$, this assumption holds for any $Gini^*$.

**Remark 2.** Compared to the Spike entropy proposed by Kirchenbauer et al. (2023a), the Gini index is a commonly used metric in machine learning to measure the uncertainty of a probability distribution, such as CART decision tree (Breiman, 2017).

**Example 1.** We show an example for a typical $\gamma$. Let $\gamma = 0.5$, this bound becomes

$$\mathbb{E}(|S|_G) \geq \frac{1}{2}T + \frac{1}{4}TGini^*, \tag{6}$$

$$\mathbb{V}(|S|_G) \leq T[\frac{1}{2} + \frac{1}{4}Gini^*][\frac{1}{2} - \frac{1}{4}Gini^*] = T[\frac{1}{4} - \frac{1}{16}Gini^{*2}]. \tag{7}$$

Note that $Gini^*$ is the average Gini index. When the generation becomes more uncertain, $Gini^*$ increases and we can expect a higher number of green list tokens with a lower variance. Practically in low-entropy scenarios, with probability masses concentrated on one or a few tokens, those tokens are likely to be generated frequently regardless of the green and red list partition in STA-1. Thus, fewer tokens in the green list are expected. This weakens the strength of watermarking methods and makes watermark detection challenging, which is consistent with the theorem.

Having established the mean and variance bounds for $|S|_G$, with an additional condition, we derive from Theorem 2 a corollary that provides an explicit upper bound on the type II error of the $z$-test in detecting STA-1.

**Corollary 1.** *Given that Theorem 2 holds, if $Gini^* > \tilde{z}/\sqrt{\gamma(1-\gamma)T}$, we have the type II error*

$$P\left(\frac{|S|_G - \gamma T}{\sqrt{\gamma(1-\gamma)T}} \leq \tilde{z}\bigg|H_a\right) \leq \frac{\overline{\mathbb{V}}}{\overline{\mathbb{V}} + (\underline{\mathbb{E}} - \gamma T - \tilde{z}\sqrt{\gamma(1-\gamma)T})^2}, \tag{8}$$

*where $\tilde{z}$ is the z threshold value, $\underline{\mathbb{E}}$ and $\overline{\mathbb{V}}$ are the lower bound and upper bound values on $\mathbb{E}(|S|_G)$ and $\mathbb{V}(|S|_G)$ as established in Theorem 2, respectively.*

*Proof. See Appendix C.3.*

The additional condition requires that $Gini^*$ must not be excessively low given the threshold value $\tilde{z}$. A higher $Gini^*$ increases $\underline{\mathbb{E}}$ and decreases $\overline{\mathbb{V}}$, resulting in a reduced upper bound on the type II error. Therefore, the test has higher statistical power in high-entropy scenarios.

### 4.1.2 DISCUSSION ON THE LOW-ENTROPY PROTOCOL

Previous work claims that unbiased watermarks can avoid the tradeoff between watermark strength and text quality (Hu et al., 2024). We challenge this claim by first considering the following example.

**Example 2.** Assuming that the token set only includes two tokens $\mathcal{V} = \{A, B\}$, at a typical step, an LLM outputs the probability of generating $A$ ($p_A$) and $B$ ($p_B$) as $(p_A, p_B) = (0.8, 0.2)$. Consider the following two unbiased watermarks. $W_1$: with a probability of 0.2 always generating $B$ and with a probability of 0.8 always generating $A$; $W_2$: with a probability of 0.5, the probability distribution becomes $(p_A, p_B) = (0.9, 0.1)$ and with the other probability of 0.5, becomes $(p_A, p_B) = (0.7, 0.3)$.

In Example 2, one can view the prompt as 'The quick brown fox jumps over the lazy', $A$ as the token 'dog', and $B$ as all other tokens. It is easy to show that watermarks $W_1$ and $W_2$ are both unbiased. However, risk-averse people (Pratt, 1978) will prefer watermark $W_2$ because $W_2$ does not have a possibility that only $B$ will be sampled. $B$ represents unsatisfactory outputs in low-entropy scenarios which could significantly harm text quality, and we want the risk of sampling $B$ to be as low as possible. We refer readers to Appendix D for a conventional example in finance and a better understanding of the analysis via utility theory. At any generation step, let $x_{max}$ denote the token with the maximum probability $p_{max}$. We measure the risk by the variance (Sharpe, 1998) of $p_{max}^{w,k}$ among watermark keys, where $p_{max}^{w,k}$ denotes the altered value of $p_{max}$ with a watermarking method and a key $k$. We show that STA-1 has a lower risk compared to previous unbiased watermarks in the following theorem.

**Theorem 3.** *Assume $1 - \alpha \leq p_{max} < 1$, where $\alpha$ represents the partition hyperparameter used in Dipmark. For the low-entropy protocol in Section 3, the STA-1 method has a lower variance in the probability of generating $x_{max}$ compared to other unbiased methods (including Dipmark, $\gamma$-reweight,*

*and RDW) (Hu et al., 2024; Wu et al., 2024; Kuditipudi et al., 2023). Formally,*

$$\mathbb{V}^{\text{STA-1}}_{k \sim P_K(k)} \left[ p^{w,k}_{max} \right] < \mathbb{V}^{\text{Dipmark}}_{k \sim P_K(k)} \left[ p^{w,k}_{max} \right] = \mathbb{V}^{\gamma\text{-reweight}}_{k \sim P_K(k)} \left[ p^{w,k}_{max} \right] < \mathbb{V}^{\text{RDW}}_{k \sim P_K(k)} \left[ p^{w,k}_{max} \right], \qquad (9)$$

*for any $\alpha \in [0, 0.5]$ used in Dipmark, where $p^{w,k}_{max}$ denotes the adjusted probability of the token $x_{max}$ under the respective watermarking method with a key $k \in K$.*

*Proof. See Appendix C.4.*

**Example 3.** We show a numerical example with $p_{max} = 0.8$. For STA-1, based on the proof of the theorem, if the proportion of the green list is 0.5, $\mathbb{V}^{\text{STA-1}}_{k \sim P_K(k)} \left[ p^{w,k}_{max} \right] = 0.0064$. For Dipmark ($\alpha \in [0.2, 0.5]$) and $\gamma$-reweight, the variance is $\mathbb{V}^{\text{Dipmark}}_{k \sim P_K(k)} \left[ p^{w,k}_{max} \right] = \mathbb{V}^{\gamma\text{-reweight}}_{k \sim P_K(k)} \left[ p^{w,k}_{max} \right] = \frac{1}{75} \approx 0.013$. For RDW, the variance is $\mathbb{V}^{\text{RDW}}_{k \sim P_K(k)} \left[ p^{w,k}_{max} \right] = 0.16$.

## 4.2 SAMPLING M THEN ACCEPTING

A low-entropy scenario indicates a low Gini index which weakens the watermark strength based on Theorem 2. To enhance the watermark strength, we propose the Sampling M Then Accepting (STA-M) method, an extension of STA-1. STA-M employs a heuristic threshold $\tau$ for entropy at each generation step. In detail, at generation step $t$, we first calculate the entropy $\tau^t$ of the probability distribution $P_M(\cdot | x^{-N_p:(t-1)})$. If it shows low entropy $\tau^t \leq \tau$, we apply STA-1 at this generation step; if it shows high entropy $\tau^t > \tau$, we repeat sampling if the previously sampled token is in the red list, and the procedure repeats at most $M$ times.

The detailed algorithm and analysis of STA-M can be found in Appendix E. According to Remark 3 in Appendix E, STA-M is biased. In low-entropy steps where probabilities are concentrated on a few tokens, actively using STA-M by repeated sampling can skew these probabilities, thereby reducing text quality. On the contrary, in high-entropy steps, since there are more acceptable tokens, the impact of repeated sampling on text quality is weakened. Therefore, STA-M only repeats sampling in high-entropy steps, which could increase watermark strength and largely maintain text quality.

## 5 EXPERIMENTS

In this section, we conducted computational experiments to evaluate the performance of STA-1 and STA-M using two public datasets. We benchmarked our methods against various watermarking baselines on text quality and watermark strength. Moreover, we discussed the variance of generation in the low-entropy dataset. Finally, we conducted a robustness analysis of STA against different watermarking attacks.

### 5.1 EXPERIMENTAL SETUP

**Datasets and metrics.** We employed two public datasets: C4 subset (Raffel et al., 2020; Kirchenbauer et al., 2023a) for news-like text generation and HumanEval (Chen et al., 2021) for code generation. We evaluated the performance of different watermarking methods on text quality and watermark strength. For watermark strength, we set the $z$ threshold as 2 and 2.5 and report the F1-score and AUC of watermark detection. For text quality, we measured perplexity (PPL) and coherence (Gao et al., 2021) for generations on C4; We computed PPL and pass@$k$ scores of code generations (Chen et al., 2021) for HumanEval. We refer readers to Appendix F.1 for more dataset details and the prompt used in each dataset.

**Baselines.** We chose KGW as the biased watermark baseline (Kirchenbauer et al., 2023a), RDW (Kuditipudi et al., 2023), $\gamma$-reweight (Hu et al., 2024), and Dipmark (Wu et al., 2024) as the unbiased watermark baselines. Specifically, we set KGW with a fixed green list proportion $\gamma = 0.5$ and diverse logit increments $\delta \in \{1, 1.5, 2\}$. We set the watermark key length as 256 in RDW. The partition parameter of Dipmark was set as $\alpha \in \{0.3, 0.4, 0.5\}$. When $\alpha = 0.5$, we report this result as $\gamma$-reweight. Note that $\gamma$-reweight (Hu et al., 2024) does not include a $z$-test. Therefore, we implemented the $z$-score in Dipmark (Wu et al., 2024) for $\gamma$-reweight by counting the number of tokens in the latter portion of the token set. We also show results without watermarking techniques. Also, RDW only contains a permutation test that reports p-values. We set p-value thresholds at 0.05 and 0.01 to approximate two $z$-tests.

Table 1: Result Comparison between Our Methods and Baselines on Text Quality and Watermark Strength for the C4 Dataset. The best results without statistical differences are shown in bold. The second best results without statistical differences are shown in underline.

| Method | Text Quality | | Watermark Strength | | | | Detection Time |
| | | | $z = 2.0$ | | $z = 2.5$ | | |
| | PPL | Coherence | F1 | AUC | F1 | AUC | |
|---|---|---|---|---|---|---|---|
| No Watermark | **7.474** | **0.604** | 0.046 | 0.500 | 0.012 | 0.500 | 46s |
| KGW($\delta$=1) | **7.591** | **0.606** | 0.961 | 0.962 | 0.940 | 0.944 | 46s |
| KGW($\delta$=1.5) | 7.844 | **0.604** | **0.985** | **0.984** | **0.992** | **0.992** | 46s |
| KGW($\delta$=2) | 8.091 | **0.599** | **0.986** | **0.986** | **0.995** | **0.995** | 46s |
| RDW | 7.650 | 0.592 | **0.982** | **0.982** | 0.948 | 0.950 | 4h |
| Dipmark($\alpha$=0.3) | **7.415** | **0.599** | 0.933 | 0.935 | 0.909 | 0.915 | 44s |
| Dipmark($\alpha$=0.4) | **7.384** | **0.601** | 0.957 | 0.957 | 0.954 | 0.955 | 44s |
| $\gamma$-reweight | **7.436** | **0.599** | 0.961 | 0.961 | 0.963 | 0.963 | 44s |
| STA-1 | **7.387** | **0.600** | 0.962 | 0.961 | 0.963 | 0.963 | 46s |
| STA-4($\tau$=1.35) | 7.711 | 0.595 | 0.973 | 0.972 | 0.988 | 0.988 | 46s |
| STA-8($\tau$=1.35) | 8.006 | 0.592 | 0.975 | 0.975 | 0.987 | 0.987 | 46s |
| STA-16($\tau$=1.35) | 8.199 | 0.588 | 0.973 | 0.972 | 0.988 | 0.988 | 46s |

Table 2: Result Comparison between Our Methods and Baselines on Text Quality and Watermark Strength for the HumanEval Dataset. The best results without statistical differences are shown in bold. The second best results without statistical differences are shown in underline.

| Method | Text Quality | | | | Watermark Strength | | | |
| | | | | | $z = 2.0$ | | $z = 2.5$ | |
| | PPL | Pass@1 | Pass@5 | Pass@10 | F1 | AUC | F1 | AUC |
|---|---|---|---|---|---|---|---|---|
| No Watermark | 3.041 | **0.138** | **0.405** | **0.537** | 0.114 | 0.494 | 0.072 | 0.497 |
| KGW($\delta$=1) | 3.078 | **0.135** | 0.326 | 0.415 | 0.471 | 0.643 | 0.416 | 0.627 |
| KGW($\delta$=1.5) | 3.499 | 0.098 | 0.308 | 0.427 | 0.720 | 0.770 | 0.650 | 0.730 |
| KGW($\delta$=2) | 3.723 | 0.098 | 0.254 | 0.372 | **0.757** | **0.795** | **0.733** | **0.785** |
| RDW | 3.159 | 0.134 | 0.362 | 0.470 | 0.408 | 0.628 | 0.343 | 0.604 |
| Dipmark($\alpha$=0.3) | 3.037 | **0.144** | **0.392** | **0.512** | 0.518 | 0.665 | 0.423 | 0.625 |
| Dipmark($\alpha$=0.4) | 3.101 | **0.141** | **0.393** | **0.512** | 0.516 | 0.668 | 0.429 | 0.634 |
| $\gamma$-reweight | 3.088 | **0.142** | **0.371** | **0.488** | 0.522 | 0.671 | 0.479 | 0.655 |
| STA-1 | 3.006 | **0.147** | **0.394** | **0.494** | 0.526 | 0.633 | 0.442 | 0.611 |
| STA-4($\tau$=1.95) | 3.175 | **0.135** | **0.392** | **0.500** | 0.633 | 0.685 | 0.594 | 0.679 |
| STA-8($\tau$=1.95) | **2.842** | **0.146** | **0.399** | **0.537** | 0.652 | 0.703 | 0.587 | 0.675 |
| STA-16($\tau$=1.95) | 3.024 | **0.140** | **0.382** | 0.476 | 0.725 | 0.764 | 0.640 | 0.717 |

**Implementation details.** We utilized different variants of LLaMA-2-7B (Touvron et al., 2023) as our generative models, and LLaMA-2-13B to compute perplexity. For hyperparameters in STA-M, we set $M \in \{4, 8, 16\}$ and two entropy thresholds $\tau$ for different datasets. We conducted a robustness check on $\tau$ in Appendix F.2 and selected different $\tau$s for different datasets in the final experiment. For each method, we run 10 times to conduct all pair-wise Tukey tests. Results in the following tables show only average values. We refer readers to Appendix F.1 for more details on implementation.

## 5.2 RESULTS ON C4

For the C4 dataset, each method generates at least 500 text sequences with at least $200 \pm 5$ tokens (Kirchenbauer et al., 2023a). Table 1 demonstrates each method's text quality, watermark strength, and detection time for 500 generations, and we present generated text examples in Appendix F.3. As depicted in Table 1, the proposed STA-1 method is efficient in detection and achieves comparable

perplexity and coherence compared to no watermark generation. The text quality results are consistent with other unbiased watermarks including RDW, Dipmark, and $\gamma$-reweight, showing STA-1 is also unbiased empirically. In terms of watermark strength, STA-M ($M \in \{4, 8, 16\}$) outperforms all unbiased watermarks and has comparable watermark strength as biased watermarks KGW ($\delta \in \{1.5, 2\}$). Overall in the high-entropy generation task, the unbiased STA-1 method is comparable to other unbiased watermarks; The STA-M method can improve the watermark strength by sacrificing minor text quality.

## 5.3 RESULTS ON HUMANEVAL

In this section, we compare our methods against baselines on the HumanEval dataset. Table 2 presents the perplexity, pass@$k$ scores, and watermark strength for all methods. Since it is better not to control the length of a code during generation, we remove detection time results. First, we focus on the result analysis for all unbiased watermarks. As reported, our STA-1 method achieves similar perplexity and pass@$k$ scores compared to no watermarking and other unbiased watermarking methods.

Table 3: Comparison on the Risk of Unsatisfactory Outputs for Unbiased Watermarks. For space concern, we denote the number of passed problems as PP, the number of passed codes as PC, and the average number of passed codes per passed problem as PC per PP (PC/PP).

| Method | PPL Variance | PP | PC | PC per PP |
|---|---|---|---|---|
| RDW | 2.202 | 77 | 219 | 2.844 |
| Dipmark($\alpha$=0.3) | 1.535 | **84** | 233 | 2.675 |
| Dipmark($\alpha$=0.4) | 1.853 | **84** | 221 | 2.631 |
| $\gamma$-reweight | 1.722 | 80 | 214 | 2.774 |
| STA-1 | **1.461** | 81 | **254** | **3.136** |

Moreover, we examine the risk of unsatisfactory outputs produced by unbiased watermarks for low-entropy generations. Specifically, we compare different unbiased watermarks in terms of four more metrics. We ran 10 times of code generation for each problem using different unbiased watermarking methods with 10 different keys. Table 3 reports the average variance of perplexity among each problem, the number of passed problems (if the problem is solved by any one generation out of 10 runs, it is considered passed), the number of passed codes, and the average number of passed codes among all passed problems. In particular, the STA-1 method demonstrates the lowest variance of perplexity compared to RDW, Dipmark($\alpha$=0.3), Dipmark($\alpha$=0.4), and $\gamma$-reweight with a variance of 1.461 compared to 2.202, 1.535, 1.853, and 1.722, respectively. A lower variance indicates a lower risk among different text generations under different keys. Additionally, we show the average number of passed codes among all passed problems. For example, 3.136 in Table 3 means among all solved problems, an average of 3.136 generated codes are accurate w.r.t. 10 generations by STA-1. We conclude from Table 3 that although Dipmark solves more problems, it fails to provide consistent accurate codes among different generations. Instead, our method outperforms other unbiased watermarks (RDW, Dipmark($\alpha$=0.3), Dipmark($\alpha$=0.4), $\gamma$-reweight) in providing consistency, with an average number of passed codes of 3.136 compared to 2.844, 2.675, 2.631, and 2.774, respectively. In summary, the STA-1 method has a lower risk when generating low-entropy texts, as discussed in Theorem 3.

In terms of watermark strength in Table 2, STA-M ($M \in \{4, 8, 16\}$) yields higher watermark strength in comparison to all unbiased watermarks while maintaining similar pass scores. The STA-16 method achieves comparable watermark strength against biased watermark KGW($\delta = 2$) with an AUC of 0.764 ($z = 2$) against 0.795. The text quality is maintained with a pass@10 of 0.476, highlighting the efficacy of the heuristics to enhance watermark strength at high-entropy generation steps.

## 5.4 ATTACKING STA

We assessed the robustness of different watermarking methods under different attacks consisting of the copy-paste attack (Kirchenbauer et al., 2023a), paraphrasing using GPT-3.5, and two configurations of the DIPPER attack (Krishna et al., 2024). Detailed settings of different attacks are described in Appendix F.4.

Table 4: Attacking Watermarks for the C4 Dataset.

| Attack Setting | No Attack | | Copy-Paste | | GPT-3.5 | | DIPPER-1 | | DIPPER-2 | |
|---|---|---|---|---|---|---|---|---|---|---|
| Method | F1 | AUC | F1 | AUC | F1 | AUC | F1 | AUC | F1 | AUC |
| KGW($\delta = 1$) | 0.96 | 0.96 | 0.68 | 0.75 | 0.27 | 0.57 | 0.13 | 0.53 | 0.15 | 0.54 |
| KGW($\delta = 1.5$) | 0.99 | 0 98 | 0.90 | 0.90 | 0.41 | 0.62 | 0.22 | 0.56 | 0.27 | 0.57 |
| KGW($\delta = 2$) | 0.99 | 0.99 | **0.95** | **0.95** | 0.54 | 0.68 | 0.30 | 0.58 | 0.40 | 0.62 |
| RDW | 0.98 | 0.98 | 0.83 | 0.79 | 0.73 | 0.78 | 0.64 | 0.73 | 0.65 | 0.73 |
| Dipmark($\alpha = 0.3$) | 0.93 | 0.94 | 0.61 | 0.70 | 0.29 | 0.57 | 0.24 | 0.55 | 0.26 | 0.55 |
| Dipmark($\alpha = 0.4$) | 0.96 | 0.96 | 0.75 | 0.79 | 0.38 | 0.61 | 0.31 | 0.58 | 0.34 | 0.59 |
| $\gamma$-reweight | 0.96 | 0.96 | 0.74 | 0.78 | 0.41 | 0.61 | 0.32 | 0.57 | 0.36 | 0.60 |
| STA-1 | 0.96 | 0.96 | 0.78 | 0.81 | 0.47 | 0.63 | 0.39 | 0.60 | 0.46 | 0.63 |
| STA-4($\tau$=1.35) | 0.97 | 0.97 | **0.95** | **0.95** | 0.72 | 0.78 | 0.65 | 0.73 | 0.69 | 0.75 |
| STA-8($\tau$=1.35) | 0.98 | 0.98 | **0.95** | **0.95** | **0.78** | **0.81** | **0.71** | **0.77** | 0.76 | 0.79 |
| STA-16($\tau$=1.35) | 0.97 | 0.97 | **0.95** | **0.95** | 0.76 | 0.80 | 0.68 | 0.74 | **0.78** | **0.81** |

Table 4 reports the F1-score and AUC of watermark detection under each attack with $z = 2$. As reported, for unbiased watermarks, RDW achieves the best result since its detection framework based on brute force search is designed to solve the robustness issue (Kuditipudi et al., 2023). In contrast, STA-M is robust against different attacks with a low detection time. For the copy-paste attack, since STA-M is based on the green-red list partition and changing a token can only affect the detection score of itself and the next token, it is naturally robust to simple text insertion and removal (Kirchenbauer et al., 2023a). Meanwhile, LLM-based attacks, such as GPT-3.5 and DIPPER, are designed to replace tokens in given texts by sampling from the LLM. STA-M effectively increases the proportion of green-list tokens by raising their probability in high-entropy scenarios without compromising too much text quality, making it difficult for LLM-based attacks to replace a substantial number of tokens in STA-M-generated text and remove the watermark. In conclusion, STA-M can generate text with high watermark strength against various attacks.

## 6 CONCLUSIONS AND FUTURE WORK

In this work, we propose a new unbiased watermarking method named STA-1. We clarify the text quality (regarding the risk of unsatisfactory outputs under the same expectation) and watermark strength tradeoff of unbiased watermarks in low-entropy scenarios. We also extend STA-1 to STA-M which can enhance watermark strength with small text quality shifts. Experimental results on low-entropy datasets prove that STA-1 is comparable to other unbiased watermarks and has a low risk. Moreover, results from the high-entropy dataset demonstrate the efficiency of STA-1 and the robustness of STA-M. Future work of our study can be conducted in several ways. First, watermarking low-entropy tasks is still challenging and future work can devise better watermarking methods. Second, future work could incorporate more datasets and generative LLMs for evaluation of our method. Third, it is also possible to consider context code history to extend the unbiased results from the token level to the sequence level.

## 7 RELATED WORK

With the development of LLMs, the idea of watermarking LLMs has been proposed (Aaronson, 2022; Kirchenbauer et al., 2023a) and widely explored. Existing white-box watermarking techniques can be categorized into watermarking during logits and probabilities generation (Kirchenbauer et al., 2023b; Lee et al., 2023; Hu et al., 2024; Wang et al., 2023; Fernandez et al., 2023; Zhao et al., 2023; Yoo et al., 2023; Ren et al., 2023; Takezawa et al., 2023), and watermarking by controlling sampling strategies (Christ et al., 2023; Kuditipudi et al., 2023; Hou et al., 2023; Fairoze et al., 2023). We refer readers to Appendix G for a detailed discussion on related work.

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

## A    RESEARCH GAP SUMMARY

Table 5: Research Gap Summary. Black-box refers to only black-box LLM needed during detection; Efficiency refers to efficient detection; Robustness refers to the robustness against simple watermarking attacks. Guarantee refers to the statistical guarantee of type II error; Unbiased refers to the same expectation requirement in low-entropy scenarios.

| Literature | Black-box | Efficiency | Robustness | Guarantee | Unbiased |
|---|---|---|---|---|---|
| Kirchenbauer et al. (2023a) | ✓ | ✓ | ✓ | ✓ | |
| Hu et al. (2024) | | ✓ | | | ✓ |
| Wu et al. (2024) | ✓ | ✓ | ✓ | | ✓ |
| Christ et al. (2023) | ✓ | ✓ | | ✓ | ✓ |
| Kuditipudi et al. (2023) | ✓ | | ✓ | | ✓ |
| Fairoze et al. (2023) | ✓ | ✓ | ✓ | | |
| STA-1 | ✓ | ✓ | ✓ | ✓ | ✓ |

## B    DETAILS OF PREVIOUS METHODS

Distribution reweighting refers to methods that adjust the output distribution $P_M(x^t|x^{-N_p:(t-1)})$ at each step $t$ by artificially increasing probabilities for certain tokens while reducing those for others. The direction and magnitude (increasing or decreasing) of change in probability mass for a token are determined by the private key $k$.

KGW (Kirchenbauer et al., 2023a) first randomly splits the vocabulary set $\mathcal{V}$ into two non-overlapping lists based on a uniformly distributed key $k$: a 'green' list and a 'red' list. This method has two versions: the 'hard' version completely ignores the red list tokens and only samples tokens from the green list; The 'soft' version adds a predefined constant $\delta$ to logits of green list tokens while keeping logits of red list tokens fixed. The soft KGW reweights distribution as

$$P_{M,w}(x^t = j|x^{-N_p:(t-1)}; k) = \frac{\exp\left(l_j^t + \mathbb{1}_{\text{Green}}(j)\delta\right)}{\sum_{i\in\text{Red}} \exp(l_i^t) + \sum_{i\in\text{Green}} \exp(l_i^t + \delta)},$$

where $j$ denotes the $j$-th token within the vocabulary set, $l_j^t$ is its logit output by the original LLM at step $t$, and $\mathbb{1}_{\text{Green}}(j)$ is an indicator function having a value of 1 when $j$ is in the green list and 0 otherwise.

Wu et al. (2024) propose an unbiased reweighting method, named Dipmark. Dipmark arranges all probability masses over the vocabulary set from the original LLM output consecutively within the interval $[0, 1]$ and then randomly permutes their orders based on a key $k$. A hyperparameter $\alpha \in [0, 0.5]$ partitions the probability interval $[0, 1]$ into three segments: $[0, \alpha]$, $(\alpha, 1 - \alpha]$, and $(1 - \alpha, 1]$. Probability masses in the first segment are set to 0, those in the second remain constant, and those in the third are doubled. Denote the token order after permutation as $\widetilde{\mathcal{V}}$, the adjusted probability for the $j$-th token within $\widetilde{\mathcal{V}}$ is $P_{M,w}(x^t = j|x^{-N_p:(t-1)}; k) = F(j|\widetilde{\mathcal{V}}) - F(j - 1|\widetilde{\mathcal{V}})$, with $F(j|\widetilde{\mathcal{V}})$ being defined as

$$F(j|\widetilde{\mathcal{V}}) = \max\left[\sum_{i\in\widetilde{\mathcal{V}}:i\leq j} P_M(x^t = i|\cdot) - \alpha, 0\right] + \max\left[\sum_{i\in\widetilde{\mathcal{V}}:i\leq j} P_M(x^t = i|\cdot) - (1 - \alpha), 0\right].$$

Another unbiased reweighting method, RDW (robust distortion-free watermark), is developed by Kuditipudi et al. (2023). We focus on the RDW method with an inverse transform sampling scheme. In RDW, the uniformly random key $k = (\Pi, u)$, where $\Pi$ represents a random shuffle of all probability masses $P_M(x^t|x^{-N_p:(t-1)})$ over the vocabulary set within the interval $[0, 1]$, and $u$ is a random value following the distribution $\text{U}(0, 1)$. RDW first permutes the order of all $P_M(x^t|x^{-N_p:(t-1)})$ within the interval $[0, 1]$ according to $\Pi$, then it utilizes $u$ as the cumulative distribution function value of $P_M(x^t|x^{-N_p:(t-1)})$ with respect to the permutation. Let $\Pi(j)$ denote the $j$-th token in the ordered

vocabulary set under the permutation $\Pi$. Following the inverse transform sampling scheme, the value $u$ is inverse transformed to generate a token through

$$x^s = \Pi(\min\{j : \sum_{i=1}^{j} P_M(x^t = \Pi(i)|x^{-N_p:(t-1)}) \geq u\}),$$

where $x^s$ is the sampled token. Therefore, we have $P_{M,w}(x^t = x^s|x^{-N_p:(t-1)}; k) = 1$, and the probabilities of all other tokens are reweighted to 0 accordingly.

## C    PROOFS

### C.1    PROOF OF THEOREM 1

To simplify notation, we denote the size of the vocabulary set $|\mathcal{V}|$ as $N$, the size of the green list as $N_G$, and the size of the red list as $N_R$. Given the proportion of green list $\gamma$, we have $N_G = \gamma N$ and $N_R = (1-\gamma)N$. At a generation step, let $p = (p_1, p_2, \cdots, p_N)$ denote the raw probability output by the LLM over the vocabulary set. Let $j$ represent a token within the vocabulary set, $j \in (1, 2, \cdots, N)$. We denote by $p_j^{w,k}$ the adjusted probability of token $j$ under the STA-1 watermarking method with key $k$. The key $k$ is sampled randomly from a uniform distribution $P_K(k)$.

To conveniently compute $\mathbb{E}_{k \sim P_K(k)}\left[p_j^{w,k}\right]$, we consider the uniformly random partition of green and red lists associated with the uniformly distributed key $k$ as the following process. Initially, token $j$ is randomly assigned to the green list with a probability of $\gamma$ and to the red list with a probability of $1-\gamma$. Subsequently, tokens are randomly sampled from the remaining pool to fill the green list, with all remaining tokens then placed in the red list. For the adjusted probability, we have

$$p_j^{w,k} = \begin{cases} p_j + \left(\sum_{i \in R} p_i\right) p_j & j \in G \\ \left(\sum_{i \in R} p_i\right) p_j & j \in R \end{cases}.$$

Next, we first analyze the scenario where $j \in G$ and compute $\mathbb{E}_{G,R:j \in G}\left[p_j^{w,k}\right]$. The expectation is taken over uniformly random partitions of green/red lists that fulfill $j \in G$. Let

$$h_j(p) = \mathbb{E}_{G,R:j \in G}\left[p_j^{w,k}\right] = \mathbb{E}_{G,R:j \in G}\left[p_j + \left(\sum_{i \in R} p_i\right) p_j\right].$$

Note that $h_j(p)$'s value remains unchanged under permutations in the order of the remaining tokens $\{p_i, i \neq j\}$. Thus, we have the equality that $h_j(p) = \mathbb{E}_\Pi[h_j(\Pi p_{-j})]$, where $\Pi$ represents a random permutation of the remaining tokens $p_{-j}$ while preserving the position of $p_j$. Since $h_j(\Pi p_{-j})$ is a linear function of $p_{-j}$, we then get

$$h_j(p) = \mathbb{E}_\Pi[h_j(\Pi p_{-j})] = h_j(\mathbb{E}_\Pi[\Pi p_{-j}]).$$

The expectation of the probability values at the remaining $(N-1)$ positions over permutations of their corresponding tokens $\mathbb{E}_\Pi[\Pi p_{-j}]$ yields a probability distribution $\bar{p}$ where $\bar{p}_j = p_j$ and $\bar{p}_i = (1-p_j)/(N-1)$ for $i \neq j$. With this $\bar{p}$, we derive that

$$h_j(p) = h_j(\bar{p}) = \mathbb{E}_{G,R:j \in G}\left[\bar{p}_j + \left(\sum_{i \in R} \bar{p}_i\right) \bar{p}_j\right]$$

$$= p_j + \frac{N_R}{N-1}(1-p_j)p_j.$$

Then, we analyze the scenario where $j \in R$ and compute $\mathbb{E}_{G,R:j \in R}\left[p_j^{w,k}\right]$. Let

$$f_j(p) = \mathbb{E}_{G,R:j \in R}\left[p_j^{w,k}\right] = \mathbb{E}_{G,R:j \in R}\left[\left(\sum_{i \in R} p_i\right) p_j\right].$$

For the same reasons as illustrated above and using the same definition of $\bar{p}$, we have

$$f_j(p) = f_j(\bar{p}) = \mathbb{E}_{G,R:j \in R}\left[\left(\sum_{i \in R} \bar{p}_i\right)\bar{p}_j\right]$$

$$= \left(p_j + \frac{(N_R - 1)(1 - p_j)}{(N - 1)}\right)p_j$$

$$= p_j^2 + \frac{(N_R - 1)}{N - 1}(1 - p_j)p_j.$$

Finally, combining the random partition process of green and red lists described at the beginning of the proof with the derived expressions for $h_j(p)$ and $f_j(p)$, we obtain that

$$\mathbb{E}_{k \sim P_K(k)}\left[p_j^{w,k}\right] = \gamma h_j(p) + (1 - \gamma)f_j(p)$$

$$= \gamma p_j + \gamma \frac{N_R}{N - 1}(1 - p_j)p_j + (1 - \gamma)p_j^2 + (1 - \gamma)\frac{(N_R - 1)}{N - 1}(1 - p_j)p_j$$

$$= \left(\gamma + \frac{N_R - (1 - \gamma)}{N - 1}\right)p_j + \left((1 - \gamma) - \frac{N_R - (1 - \gamma)}{N - 1}\right)p_j^2$$

$$= p_j,$$

with $N_R = (1 - \gamma)N$. This concludes the proof.

## C.2 Proof of Theorem 2

In this proof, we employ the notations introduced in the proof of Theorem 1 in Section C.1, and we leverage the results derived from that theorem's proof.

For a token $j$ within the vocabulary set, $j \in (1, 2, \cdots, N)$, we consider the identical random partition process of green and red lists as described at the beginning of the proof of Theorem 1. If $j$ is initially assigned to the green list, according to the proof of Theorem 1, its expected adjusted probability over uniformly random green/red list partitions that fulfill $j \in G$ satisfies

$$\mathbb{E}_{G,R:j \in G}\left[p_j^{w,k}\right] = p_j + \frac{N_R}{N - 1}(1 - p_j)p_j$$

$$= p_j\left[\frac{N - 1 + N_R(1 - p_j)}{N - 1}\right]$$

$$\geq p_j\left[\frac{N + N_R(1 - p_j)}{N}\right]$$

$$= p_j + (1 - \gamma)p_j(1 - p_j),$$

where the inequality holds because the denominator is less than the numerator, and adding 1 to both leads to a decrease in the value.

Recall that each token within the vocabulary set has a probability of $\gamma$ being assigned to the green list. Thus, the overall probability of sampling a token from the green list has the lower bound

$$\mathbb{P}(G) := \mathbb{P}(\text{sampling a token} \in G) = \sum_{j=1}^{N} \gamma \mathbb{E}_{G,R:j \in G}\left[p_j^{w,k}\right]$$

$$\geq \gamma \sum_{j=1}^{N} p_j + (1 - \gamma)p_j(1 - p_j)$$

$$= \gamma + \gamma(1 - \gamma)\sum_{j=1}^{N} p_j(1 - p_j).$$

Note that this lower bound applies to every generation step $t$. Let $p^t$ denote the LLM's original output probability distribution at step $t$, and $G^t$ denote the event of sampling a token from the green list at

step $t$, we then have

$$\mathbb{P}(G^t) \geq \gamma + \gamma(1-\gamma)\sum_{j=1}^{N} p_j^t(1-p_j^t) = \gamma + \gamma(1-\gamma)Gini(p^t).$$

It is important to highlight that this lower bound holds significant meaning, as it strictly exceeds the naive lower bound for $\mathbb{P}(G^t)$, which is $\gamma$. This bound serves as a crucial element in the proof of Theorem 2. For the expectation of the number of green list tokens in the sequence, we can derive that

$$\mathbb{E}(|S|_G) = T\mathbb{E}_t\left[\mathbb{P}(G^t)\right] \geq T\mathbb{E}_t\left[\gamma + \gamma(1-\gamma)Gini(p^t)\right]$$
$$\geq T\left[\gamma + \gamma(1-\gamma)Gini^*\right] = \gamma T + (1-\gamma)\gamma T Gini^*,$$

where the lower bound $Gini^*$ for the average Gini index is provided as a condition in the theorem.

Next, regarding the variance of $|S|_G$, it is worth noting that the success of sampling a token from the green list at each step $t$ can be viewed as a Bernoulli random variable with a success probability of $\mathbb{P}(G^t)$. This Bernoulli random variable has a variance of $\mathbb{P}(G^t)[1 - \mathbb{P}(G^t)]$. The sum of these Bernoulli random variables across all $T$ steps gives us $|S|_G$. Because these random variables are independent of each other, the variance of their sum equals the sum of their variances. Consequently, we can obtain that

$$\mathbb{V}(|S|_G) = T\mathbb{E}_t\left[\mathbb{P}(G^t)[1 - \mathbb{P}(G^t)]\right]$$
$$\leq T\mathbb{E}_t[\mathbb{P}(G^t)]\left[1 - \mathbb{E}_t[\mathbb{P}(G^t)]\right]$$
$$\leq T\left[\gamma + (1-\gamma)\gamma Gini^*\right]\left[1 - \gamma - (1-\gamma)\gamma Gini^*\right],$$

where the first inequality holds by applying Jensen's inequality to a concave function of $\mathbb{P}(G^t)$, and the second inequality is valid because 1) $\mathbb{E}_t\left[\mathbb{P}(G^t)\right] \geq \gamma + (1-\gamma)\gamma Gini^*$ as shown above; 2) the function $x(1-x)$ is decreasing in the range $x \in [0.5, 1]$; and 3) it is assumed in the theorem that $\gamma + (1-\gamma)\gamma Gini^* \geq 0.5$. This concludes the proof.

## C.3 PROOF OF COROLLARY 1

For the $z$-test in detecting STA-1, its type II error is defined as $P(z \leq \tilde{z}|H_a)$. Following the definition, we have that

$$P(z \leq \tilde{z}|H_a) = P\left(\frac{|S|_G - \gamma T}{\sqrt{\gamma(1-\gamma)T}} \leq \tilde{z}\,\middle|\,H_a\right)$$
$$= P(|S|_G - \mathbb{E}(|S|_G) \leq \gamma T + \tilde{z}\sqrt{\gamma(1-\gamma)T} - \mathbb{E}(|S|_G)|H_a)$$
$$\leq P(|S|_G - \mathbb{E}(|S|_G) \leq \gamma T + \tilde{z}\sqrt{\gamma(1-\gamma)T} - \underline{\mathbb{E}}|H_a)$$
$$\leq \frac{\mathbb{V}(|S|_G)}{\mathbb{V}(|S|_G) + (\underline{\mathbb{E}} - (\gamma T + \tilde{z}\sqrt{\gamma(1-\gamma)T}))^2} \qquad \text{(Cantelli's inequality)}$$
$$\leq \frac{\overline{\mathbb{V}}}{\overline{\mathbb{V}} + (\underline{\mathbb{E}} - \gamma T - \tilde{z}\sqrt{\gamma(1-\gamma)T})^2},$$

where Cantelli's inequality holds because

$$\underline{\mathbb{E}} - (\gamma T + \tilde{z}\sqrt{\gamma(1-\gamma)T}) = \gamma(1-\gamma)T Gini^* - \tilde{z}\sqrt{\gamma(1-\gamma)T} > 0$$

according to the condition assumed in the corollary. This completes the proof.

## C.4 PROOF OF THEOREM 3

In this proof, we continue utilizing the notations introduced in the proof of Theorem 1 in Section C.1.

We start with the variance calculation for the STA-1 method. Because STA-1 is an unbiased watermark by Theorem 1, we have $\mathbb{V}_{k\sim P_K(k)}^{\text{STA-1}}\left[p_{max}^{w,k}\right] = \mathbb{E}_{k\sim P_K(k)}^{\text{STA-1}}\left[(p_{max}^{w,k} - p_{max})^2\right]$. Considering the identical uniformly random partition process of green and red lists associated with the uniformly

distributed key $k$ as in the proof of Theorem 1, depending on whether the token $x_{max}$ is assigned to the green list or not initially, $p_{max}^{w,k}$ have two possible realizations:

$$p_{max}^{w,k} = \begin{cases} p_{max} + \left(\sum_{i \in R} p_i\right) p_{max} & x_{max} \in G \\ \left(\sum_{i \in R} p_i\right) p_{max} & x_{max} \in R \end{cases}.$$

Under the assumption that the probabilities of the other $N-1$ tokens uniformly fill in the remaining $(1 - p_{max})$ probability mass, each $p_i$, $i \in (1, 2, \cdots, N)$ and $i \neq x_{max}$, equals $(1 - p_{max})/(N - 1)$. Therefore, if $x_{max} \in G$, $p_{max}^{w,k} = p_{max} + N_R(1 - p_{max})p_{max}/(N - 1)$, and this value is fixed for all partitions of green/red lists that fulfill $x_{max} \in G$. Then we have

$$\mathbb{E}_{G,R:x_{max} \in G}^{\text{STA-1}} \left[(p_{max}^{w,k} - p_{max})^2\right] = \left[\frac{N_R(1 - p_{max})p_{max}}{(N - 1)}\right]^2.$$

Similarly, if $x_{max} \in R$, we get

$$\mathbb{E}_{G,R:x_{max} \in R}^{\text{STA-1}} \left[(p_{max}^{w,k} - p_{max})^2\right] = \left[\left(\frac{(N_R - 1)(1 - p_{max})}{N - 1} + p_{max}\right) p_{max} - p_{max}\right]^2.$$

With these two expected values, and recalling that $x_{max}$ has a probability of $\gamma$ of being assigned to the green list and a probability of $1 - \gamma$ of being assigned to the red list, the variance for the STA-1 method is

$$\mathbb{V}_{k \sim P_K(k)}^{\text{STA-1}} \left[p_{max}^{w,k}\right] = \mathbb{E}_{k \sim P_K(k)}^{\text{STA-1}} \left[(p_{max}^{w,k} - p_{max})^2\right]$$

$$= \gamma \mathbb{E}_{G,R:x_{max} \in G}^{\text{STA-1}} \left[(p_{max}^{w,k} - p_{max})^2\right] + (1 - \gamma) \mathbb{E}_{G,R:x_{max} \in R}^{\text{STA-1}} \left[(p_{max}^{w,k} - p_{max})^2\right]$$

$$= \gamma \left[\frac{N_R(1 - p_{max})p_{max}}{(N - 1)}\right]^2 + (1 - \gamma) \left[\left(\frac{(N_R - 1)(1 - p_{max})}{N - 1} + p_{max}\right) p_{max} - p_{max}\right]^2$$

$$= p_{max}^2 (1 - p_{max})^2 \left[\gamma \frac{N_R^2}{(N - 1)^2} + (1 - \gamma) \frac{N_G^2}{(N - 1)^2}\right]$$

$$= p_{max}^2 (1 - p_{max})^2 \gamma (1 - \gamma) \frac{N^2}{(N - 1)^2}.$$

Next, we compute the variance for the Dipmark method with a partition hyperparameter $\alpha$. Note that $\mathbb{V}_{k \sim P_K(k)}^{\text{Dipmark}} \left[p_{max}^{w,k}\right] = \mathbb{E}_{k \sim P_K(k)}^{\text{Dipmark}} \left[(p_{max}^{w,k} - p_{max})^2\right]$ holds because Dipmark is also unbiased. In Dipmark, the uniformly distributed key $k$ controls the randomness of permutations. Under the same assumption that $p_i = (1 - p_{max})/(N - 1)$ for $i \neq x_{max}$, the relative orders among these $(N - 1)$ tokens become irrelevant in the permutation. Therefore, there are a total of $N$ unique permutations, each with a probability of $1/N$. Specifically, in the first unique permutation, there are 0 tokens $i$ where $i \neq x_{max}$ placed to the left of $x_{max}$ and $(N - 1)$ tokens $i$ where $i \neq x_{max}$ placed to the right of $x_{max}$. In the second one, there is 1 token on the left and $(N - 2)$ tokens on the right, and so forth. The last permutation has $(N - 1)$ tokens on the left and 0 on the right. If $j$ such tokens are on the left of $x_{max}$, $j = 0, 1, \cdots, (N - 1)$, the corresponding $p_{max}^{w,k}$ is

$$p_{max}^{w,k} = 2p_{max} - 1 + 2j \frac{(1 - p_{max})}{(N - 1)},$$

given that $1 - \alpha \leq p_{max} < 1$ as assumed in the condition. Therefore, the variance for the Dipmark method with a partition hyperparameter $\alpha$ is

$$\mathbb{V}_{k \sim P_K(k)}^{\text{Dipmark}} \left[p_{max}^{w,k}\right] = \mathbb{E}_{k \sim P_K(k)}^{\text{Dipmark}} \left[(p_{max}^{w,k} - p_{max})^2\right]$$

$$= \frac{1}{N} \sum_{j=0}^{N-1} \left[p_{max} - 1 + 2j \frac{(1 - p_{max})}{(N - 1)}\right]^2$$

$$= -(p_{max} - 1)^2 + \frac{1}{N} \sum_{j=0}^{N-1} 4j^2 \frac{(1 - p_{max})^2}{(N - 1)^2}$$

$$= (1 - p_{max})^2 \frac{(N + 1)}{3(N - 1)}.$$

Note that, this variance value does not depend on $\alpha$. When $\alpha = 0.5$, Dipmark becomes $\gamma$-reweight. Therefore, $\mathbb{V}^{\text{Dipmark}}_{k \sim P_K(k)} \left[ p^{w,k}_{max} \right] = \mathbb{V}^{\gamma\text{-reweight}}_{k \sim P_K(k)} \left[ p^{w,k}_{max} \right]$.

Finally, we determine the variance for the RDW method with an inverse transform sampling scheme. In RDW, the uniformly distributed key $k = (\Pi, u)$, where $\Pi$ is a uniformly random permutation of the $N$ tokens and $u \sim U(0, 1)$. Similar to the previous analysis of Dipmark, the relative orders among the remaining $(N - 1)$ tokens except $x_{max}$ are irrelevant to the permutation. Therefore, we only need to consider the $N$ unique permutations, each with a probability of $1/N$, as discussed above. Conditional on any permutation $\Pi$, under the inverse transform sampling scheme, there is a probability of $p_{max}$ that $x_{max}$ will be sampled out. Therefore, the altered value of $p_{max}$ given $\Pi$ is

$$p^{w,k}_{max} | \Pi = \begin{cases} 1 & \text{with probability } p_{max} \\ 0 & \text{with probability } 1 - p_{max} \end{cases}.$$

Then, we have that
$$\mathbb{V}^{\text{RDW}}_u \left[ p^{w,k}_{max} | \Pi \right] = p_{max}(1 - p_{max}).$$
Because these results hold for any permutation $\Pi$, by the law of total variance, we can derive that

$$\mathbb{V}^{\text{RDW}}_{k \sim P_K(k)} \left[ p^{w,k}_{max} \right] = \mathbb{E}_\Pi \left( \mathbb{V}^{\text{RDW}}_u \left[ p^{w,k}_{max} | \Pi \right] \right) + \mathbb{V}_\Pi \left( \mathbb{E}^{\text{RDW}}_u \left[ p^{w,k}_{max} | \Pi \right] \right)$$
$$= p_{max}(1 - p_{max}) + 0$$
$$= p_{max}(1 - p_{max}),$$

which is the variance for the RDW method with an inverse transform sampling scheme.

To compare $\mathbb{V}^{\text{STA-1}}_{k \sim P_K(k)} \left[ p^{w,k}_{max} \right]$ and $\mathbb{V}^{\text{Dipmark}}_{k \sim P_K(k)} \left[ p^{w,k}_{max} \right]$, consider

$$\mathbb{V}^{\text{STA-1}}_{k \sim P_K(k)} \left[ p^{w,k}_{max} \right] = p^2_{max}(1 - p_{max})^2 \gamma(1 - \gamma) \frac{N^2}{(N-1)^2}$$
$$< \frac{1}{4}(1 - p_{max})^2 \frac{N^2}{(N-1)^2}$$
$$= (1 - p_{max})^2 \frac{(N+1)}{3(N-1)} \times \frac{3}{4} \frac{N^2}{N^2 - 1},$$

where $N^2/(N^2 - 1)$ is a decreasing function on $N$ and $N^2/(N^2 - 1) < 4/3$ for $N > 2$. Therefore, for a real-world vocabulary set where $N \gg 2$, we have

$$\mathbb{V}^{\text{STA-1}}_{k \sim P_K(k)} \left[ p^{w,k}_{max} \right] < (1 - p_{max})^2 \frac{(N+1)}{3(N-1)} = \mathbb{V}^{\text{Dipmark}}_{k \sim P_K(k)} \left[ p^{w,k}_{max} \right].$$

For the comparison between $\mathbb{V}^{\text{Dipmark}}_{k \sim P_K(k)} \left[ p^{w,k}_{max} \right]$ and $\mathbb{V}^{\text{RDW}}_{k \sim P_K(k)} \left[ p^{w,k}_{max} \right]$, we have that

$$\mathbb{V}^{\text{Dipmark}}_{k \sim P_K(k)} \left[ p^{w,k}_{max} \right] = (1 - p_{max})^2 \frac{(N+1)}{3(N-1)}$$
$$< (1 - p_{max})^2$$
$$\leq p_{max}(1 - p_{max}) = \mathbb{V}^{\text{RDW}}_{k \sim P_K(k)} \left[ p^{w,k}_{max} \right],$$

where the first inequality holds because $(N + 1) < 3(N - 1)$ for $N > 2$, and the second inequality is valid under the assumption that $1 - \alpha \leq p_{max} < 1$ and $\alpha \in [0, 0.5]$.

Putting all the results together, we get

$$\mathbb{V}^{\text{STA-1}}_{k \sim P_K(k)} \left[ p^{w,k}_{max} \right] < \mathbb{V}^{\text{Dipmark}}_{k \sim P_K(k)} \left[ p^{w,k}_{max} \right] = \mathbb{V}^{\gamma\text{-reweight}}_{k \sim P_K(k)} \left[ p^{w,k}_{max} \right] < \mathbb{V}^{\text{RDW}}_{k \sim P_K(k)} \left[ p^{w,k}_{max} \right],$$

which concludes the proof.

## D  EXAMPLE OF RISK-AVERSE

**St. Petersburg paradox (Wikipedia, 2024).** Assume that one must choose either one lottery from the following two lotteries. (1) Lottery 1 ($L1$) has a 0.8 probability of earning nothing and the other

0.2 probability of losing 1,000 dollars. (2) Lottery 2 ($L2$) has a 0.5 probability of losing 100 dollars and the other 0.5 probability of losing 300 dollars.

It is easy to show that $L1$ and $L2$ have the same expected outcome that $0.8 \times 0 - 0.2 \times 1000 = -0.5 \times 100 - 0.5 \times 300 = -200$. However, risk-averse people will choose $L2$ as they do not want to take the risk of losing 1,000 dollars.

Computationally, assume the person has 1,001 dollars in total and the utility function is $\ln(Y)$ (Debreu et al., 1954), where $Y$ is the wealth. The utility function measures happiness. It is a concave function (such as $\ln(Y)$) because people are happier if they are wealthier ($\ln'(Y) > 0$) but the increment of happiness decreases as the wealth increases ($\ln''(Y) < 0$).

The weighted utility of $L1$ and $L2$ are as follows

$$U(L1) = 0.8 \times \ln(1001) + 0.2 \times \ln(1) \approx 5.53,$$

$$U(L2) = 0.5 \times \ln(901) + 0.5 \times \ln(701) \approx 6.68.$$

Based on the weighted utility, risk-averse people will choose $L2$.

Link the lottery example to Example 2 in Section 4.1.2. Because of the low-entropy setting, sampling $B$ results in a huge loss in text quality. Suppose we treat sampling $A$ as earning nothing and sampling $B$ as losing 1,000 for text quality. In this case, we should minimize the risk of sampling $B$. Also in this case, the two unbiased watermarks in Example 2 can be viewed as $L1$ and $L2$ in the lottery example. Sampling $B$ may not be a big issue in high-entropy scenarios because it should not significantly harm text quality as much as 1,000.

---

**Algorithm 2** STA-M Text Generation

---

**Input:** A pretrained LLM $P_M$, a key $k \in K$, the proportion of green list $\gamma \in (0, 1)$, the number of maximum samples per step $M$, a entropy threshold $\tau$, and a prompt $x^{-N_p:0}$

1: **for** $t = 1, 2, \ldots, T$ **do**
2:     Get the probability distribution of tokens $p^t = P_M(\cdot | x^{-N_p:(t-1)})$
3:     Compute the entropy $\tau^t$ of $p^t$
4:     **if** $\tau^t < \tau$ **then**
5:        $M^t = 1$
6:     **else**
7:        $M^t = M$
8:     **end if**
9:     Compute the hash of the last token $x^{t-1}$. Partition the token set $\mathcal{V}$ to form the green $G$ and red $R$ list based on key $k$, the hash, and the proportion $\gamma$
10:    Initialize sample number $m = 1$
11:    **while** $m \le M^t$ and the next token $x^t$ not defined **do**
12:       Sample the candidate token $x_{c,m}^t$ with $p^t$
13:       **if** $x_{c,m}^t \in G$ **then**
14:          Accept the sampling, the next generated token $x^t = x_{c,m}^t$
15:       **else**
16:          $m \leftarrow m + 1$
17:       **end if**
18:    **end while**
19:    **if** the next token $x^t$ not defined **then**
20:       Sample $x^t$ from the distribution $p^t$
21:    **end if**
22: **end for**

**Output:** The generated text $x^{1:T}$

---

# E   STA-M DETAILS

The detailed algorithm of STA-M is shown in Algorithm 2.

**Remark 3.** STA-M is not unbiased.

We provide a counterexample to show that STA-M is biased. Assume that the vocabulary set consists of four tokens $\{a, b, c, d\}$, and at a generation step, the raw probabilities output by the LLM for these tokens are $\{p_a = 1/2, p_b = 1/3, p_c = p_d = 1/12\}$. The proportion of green list $\gamma$ equals 0.5. Therefore, with a key $k$, two tokens are randomly assigned to the green list, and the red list contains the other two. For the uniformly distributed key $k$, there are six possible random partitions of green and red lists: $\{a, b \in G; c, d \in R\}$, $\{a, c \in G; b, d \in R\}$, $\{a, d \in G; b, c \in R\}$, $\{b, c \in G; a, d \in R\}$, $\{b, d \in G; a, c \in R\}$, and $\{c, d \in G; a, b \in R\}$, each with a probability of $1/6$. Next, considering the token $a$, its adjusted probability under the STA-M watermarking method for each of the six partitions is:

$$
p_a^{w,k} = \begin{cases}
\frac{1}{2} + \frac{1}{6} \times \frac{1}{2} + (\frac{1}{6})^2 \times \frac{1}{2} + \cdots + (\frac{1}{6})^M \times \frac{1}{2} & \{a, b \in G; c, d \in R\} \\
\frac{1}{2} + \frac{5}{12} \times \frac{1}{2} + (\frac{5}{12})^2 \times \frac{1}{2} + \cdots + (\frac{5}{12})^M \times \frac{1}{2} & \{a, c \in G; b, d \in R\} \\
\frac{1}{2} + \frac{5}{12} \times \frac{1}{2} + (\frac{5}{12})^2 \times \frac{1}{2} + \cdots + (\frac{5}{12})^M \times \frac{1}{2} & \{a, d \in G; b, c \in R\} \\
(\frac{7}{12})^M \times \frac{1}{2} & \{b, c \in G; a, d \in R\} \\
(\frac{7}{12})^M \times \frac{1}{2} & \{b, d \in G; a, c \in R\} \\
(\frac{5}{6})^M \times \frac{1}{2} & \{c, d \in G; a, b \in R\}
\end{cases}.
$$

With these adjusted probability values, the expectation of the adjusted probability over the six possible partitions is easily derived as

$$
\mathbb{E}_{k \sim P_K(k)} \left[ p_a^{w,k} \right] = \frac{1}{12} \left[ \frac{6}{5} \left( 1 - (\frac{1}{6})^{M+1} \right) + 2 \times \frac{12}{7} \left( 1 - (\frac{5}{12})^{M+1} \right) + 2 \times (\frac{7}{12})^M + (\frac{5}{6})^M \right]
$$

$$
= \frac{27}{70} - \frac{1}{10}(\frac{1}{6})^{M+1} - \frac{2}{7}(\frac{5}{12})^{M+1} + \frac{1}{6}(\frac{7}{12})^M + \frac{1}{12}(\frac{5}{6})^M,
$$

which equals $p_a = 1/2$ only when $M = 1$ and is less than $1/2$ for $M \geq 2$. Hence, this counterexample demonstrates that the STA-M method is biased.

# F EXPERIMENT

## F.1 EXPERIMENTAL SETUP

**Datasets and metrics.** We employed two public datasets which are C4 subset (Raffel et al., 2020; Kirchenbauer et al., 2023a) for news-like text generation and HumanEval (Chen et al., 2021) for code generation. Specifically, C4 represents the high-entropy generation task and HumanEval represents the low-entropy generation task.

**C4:** We extracted random text segments from the news-like subset of the C4 dataset (Raffel et al., 2020) following Kirchenbauer et al. (2023a). For each segment, we removed a fixed number of tokens from the end and the removed tokens served as a 'baseline' completion. The remaining tokens were used as the prompt.

**HumanEval:** HumanEval includes 164 Python problems with test cases and solutions written by humans. We prompted the LLM with these problems. In particular, the prompt was devised as 'Below is an instruction that describes a task. Write a response that appropriately completes the request. ### Instruction: Complete the following Python code without any tests or explanation [INPUT] ### Response:'.

We evaluated the performance of different watermarks on text quality and watermark strength. For watermark strength, we implemented the $z$-test for all baselines and our methods. We set the $z$ threshold as 2 and 2.5. With $z \geq 2$, we are more than 97.7% confident that the text is watermarked based on the one-tail test.

For text quality, we employed different metrics for different datasets. For the C4 dataset, we utilized perplexity (PPL) and coherence (Gao et al., 2021) to measure the text quality. For HumanEval, we employed PPL and pass@$k$ score of the code (Chen et al., 2021). The pass@$k$ score measures the normalized percentage of solved problems in HumanEval. Formally, the pass score is calculated as

$$
\text{pass@}k = \mathbb{E}_{\text{Problems}} \left[ 1 - \frac{C_{n-c}^k}{C_n^k} \right],
$$

where $c$ is the number of passed codes among $k$ generations.

**Baselines.** We compared against biased and unbiased watermarks in terms of text quality and watermark strength. For further details of baselines, we refer readers to Appendix B. We implemented all LLMs with the Hugging Face library (Wolf et al., 2019). All watermark benchmarks including KGW, RDW, $\gamma$-reweight, and Dipmark were implemented using their public codes.

**Implementation details.** For all baselines and our methods, we utilized multinomial sampling during text generation. For C4, we employed LLaMA-2-7B as our generative LLM (Touvron et al., 2023). Following previous work (Kirchenbauer et al., 2023a), we continued to sample prompts from C4 until we had generated at least 500 text sequences, each consisting of $T = 200 \pm 5$ tokens. We leveraged LLaMA-2-13B to compute the perplexity of the generated texts. For HumanEval, we applied CodeLLaMA-7B-Instruct (Roziere et al., 2023) as the generative LLM to generate codes for all Python problems. We also leveraged LLaMA-2-13B to compute the perplexity. All experiments were conducted on a single Nvidia A100 GPU with 80GB memory.

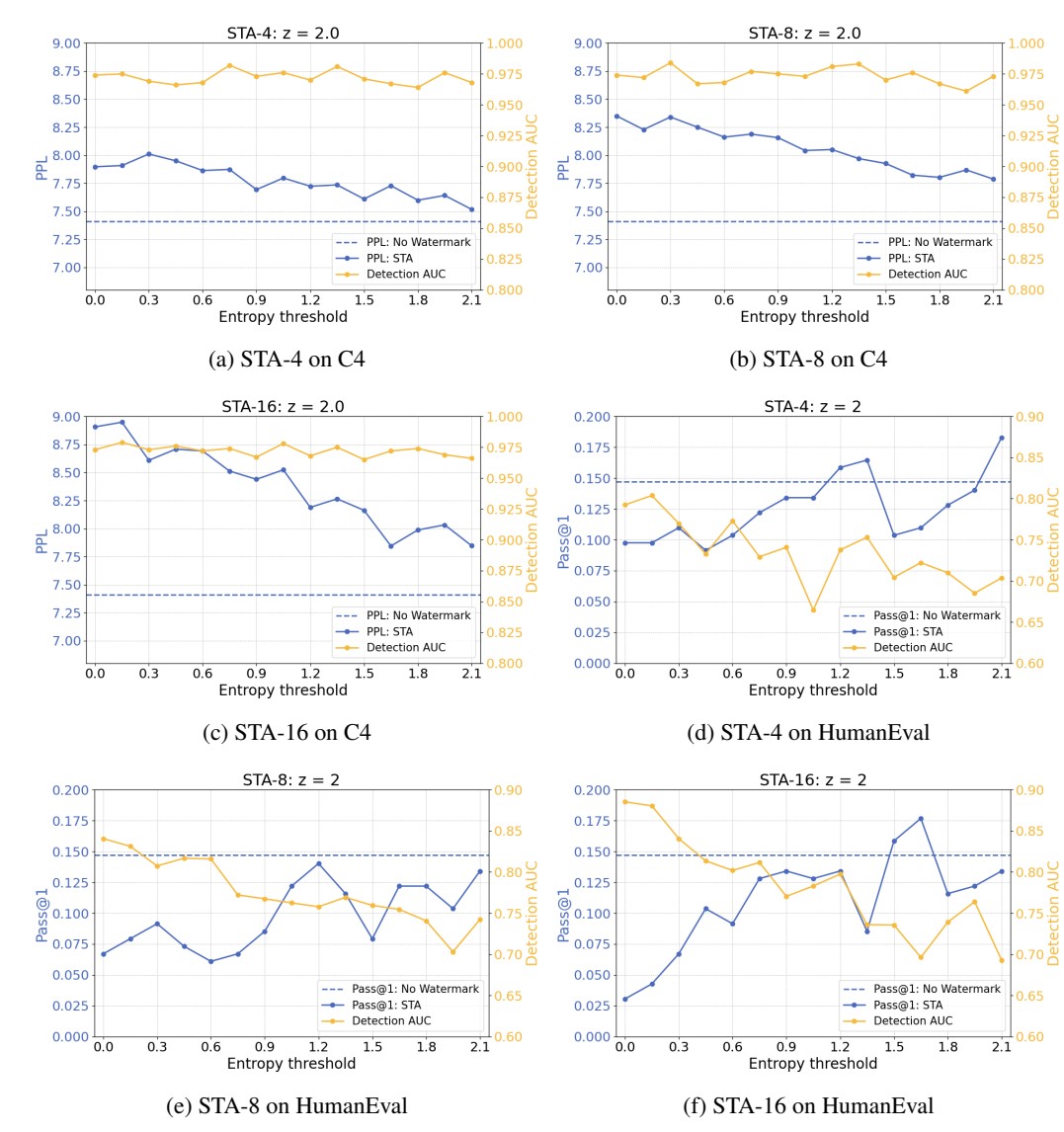

(a) STA-4 on C4

(b) STA-8 on C4

(c) STA-16 on C4

(d) STA-4 on HumanEval

(e) STA-8 on HumanEval

(f) STA-16 on HumanEval

Figure 1: Performance of STA-M w.r.t. $\tau$

## F.2 ROBUSTNESS CHECK ON ENTROPY THRESHOLD PARAMETER

In this section, we conducted a robustness check on the parameter $\tau$ in STA-M. In particular, we set the low entropy threshold $\tau$ from 0 to 2.1 with an interval of 0.15. At each generation step, we

Table 6: Examples of STA-generated Texts for C4

| Prompt | Human-written | STA-1 generated | STA-16 generated |
|---|---|---|---|
| [. . . ] Single taxpayers who are eligible to participate in a workplace retirement plan are also eligible to make a tax-deductible contribution to an IRA if their adjusted gross income is below $64,000 ($103,000 for marrieds) in 2019. This is up from $63,000 (singles) and $101,000 (marrieds) in 2018. This deduction is phased out when AGI is between $64,000 | and $74,000 (singles) and $103,000 to $123,000 (marrieds). The income range for making contributions to a Roth IRA in 2019 is $122,000 to $137,00 (singles and heads of households) and $193,000 to $203,000 (marrieds). The 2019 income limit for the Savers Credit (also called the retirement savings contributions tax credit), which is for low- to middle-income workers who contribute to a retirement plan or IRA, [. . . ] | (PPL:3.09) and $74,000 for singles ($103,000 and $123,000 for marrieds, respectively). IRA contributions can be made until the 2018 tax-return deadline, April 15, 2018, for those that filed an extension. However, you'll need to make these contributions with the 2018 deduction in mind. This means you must make IRA contributions no later than Dec. 31, 2018, to benefit on your 2018 return. [. . . ] | (PPL:3.11) and $74,000 ($103,000 and $123,000 for marrieds) and fully eliminated when AGI exceeds $74,000 (marrieds phase out at $123,000). If you're not able to participate in a 401(k) or other workplace retirement plan, you may qualify to deduct your total IRA contributions even if your income exceeds certain amount if you meet certain conditions (a deductible contributions means you won't owe tax on the contributions). [. . . ] |
| [. . . ] Thomas will be responsible for overseeing Micron's solid state storage business that ranges from hard disk drive replacements with solid state drives (SSDs) to enterprise-class storage solutions. He brings more than 30 years of experience to Micron and most recently served as the vice president of Enterprise Storage for | Micron's common stock is traded on the NASDAQ under the MU symbol. To learn more about Micron Technology, Inc., visit www.micron.com. Micron and the Micron orbit logo are trademarks of Micron Technology, Inc. All other trademarks are the property of their respective owners. [. . . ] | (PPL:3.25) the Americas region for Seagate Technology. He is a senior executive level leader with a proven track record in defining strategy that drives revenue, profit and new technology execution. "Micron is thrilled to have Darren as part of our team," said Mary Jane Raymond, . [. . . ] | (PPL:4.45) Fusion I/O, LLC. Before that, Thomas was at Western Digital Corporation where he was a progressive executive, holding various management roles since 2008, most recently as its executive vice president of storage technology. [. . . ] |
| [. . . ] Sanabia has benefited from the two times Miami's offense has given its starters decent run support, including his last outing against Washington. The 24-year-old allowed two runs and six hits over six innings in Tuesday's 8-2 victory over the Nationals. He tossed six scoreless frames in | his only road start against the New York Mets, but is allowing left-handed hitters to bat 8-for-24 against him - a troubling trend against a Reds team that features Choo, Votto and Jay Bruce at the top of the order. [. . . ] | (PPL:4.30) his prior start at Colorado. Sanoobia is 3-4 with a 4.53 ERA in 13 starts for the Marlins, who are off to the second-worst start in franchise history at 5-13. Johnny Cueto (2-3, 2.63 ERA) was hit around for five earned runs over 6 2/3 innings in a loss to Colorado last Saturday. [. . . ] | (PPL:5.30) a 5-1 home loss to the L.A. Dodgers eight days earlier. Reds rookie Anthony DeSclafani produced an excellent performance the last time he stepped onto Great American Ball Park. The young right-hander used excellent command of his off-speed pitches to strike out eight [. . . ] |

sampled at most 4, 8, and 16 times (i.e., STA-4, STA-8, and STA-16) when the entropy was above the threshold $\tau$. Figure 1 shows text quality and watermark strength of STA-M with different $\tau$s. As depicted, different $\tau$s do not affect the watermark strength significantly for C4 because C4 is a high-entropy dataset. Also, we observe a decrease in PPL when we increase $\tau$ in Figure 1a, 1b, and 1c. The reason is that by setting up a higher entropy threshold, fewer generation steps will apply the STA-M strategy, making the watermarking method more similar to STA-1. According to Figure 1d, 1e, and 1f, we observe a general increase of watermark strength if we have a larger $\tau$ because we will have more green list tokens if we sample M times instead of once. However, higher watermark strength leads to a lower pass@1 score, which is related to the text quality (Kirchenbauer et al., 2023a). We chose the Pareto optimal of each dataset as our final parameter for each dataset. Specifically, we selected $\tau = 1.35$ for C4 and $\tau = 1.95$ for HumanEval.

### F.3 EXAMPLES OF STA-GENERATED TEXTS

We present examples of STA-generated texts for C4 and HumanEval in Table 6 and Table 7, respectively. Also, we report the PPL of the generated text, and whether the code is passed specifically for HumanEval.

### F.4 ATTACKING WATERMARKS

We introduce the implementation of different attacks as follows. For the copy-paste attack, we randomly replaced $25\%$ of tokens in the watermarked text with tokens from non-watermarked text generated from the same prompt (Kirchenbauer et al., 2023a). For the GPT-3.5 attack, we utilized the prompt 'Rewrite the following paragraph: [INPUT]' for GPT-3.5. For DIPPER-1 (Krishna

Table 7: Examples of STA-generated Texts for HumanEval

| Prompt | Canonical Solution | STA-1 generated | STA-16 generated |
|---|---|---|---|
| from typing import List def filter_by_prefix (strings: List[str], prefix: str) -> List[str]: """ Filter an input list of strings only for ones that start with a given prefix. >>> filter_by_prefix([], 'a') [] >>> filter_by_prefix(['abc', 'bcd', 'cde', 'array'], 'a') ['abc', 'array'] """ | return [x for x in strings if x.startswith(prefix)] | (Pass, PPL:1.92) from typing import List def filter_by_prefix (strings: List[str], prefix: str) -> List[str]: """ Filter an input list of strings only for ones that start with a given prefix. >>> filter_by_prefix([], 'a') [] >>> filter_by_prefix(['abc', 'bcd', 'cde', 'array'], 'a') ['array', 'abc'] """ return [word for word in strings if word.startswith(prefix)] | (Pass, PPL:2.18) from typing import List def filter_by_prefix (strings: List[str], prefix: str) -> List[str]: """* Filter an input list of strings only for ones that start with a given prefix. Input: 'strings = ["abc", "bcd", "cde", "array"]', 'prefix = 'a'' Output: '["abc", "array"]' """ return [word for word in strings if word.startswith(prefix)] |
| from typing import List def intersperse (numbers: List[int], delimeter: int) -> List[int]: """ Insert a number 'delimeter' between every two consecutive elements of input list 'numbers' >>> intersperse([], 4) [] >>> intersperse([1, 2, 3], 4) [1, 4, 2, 4, 3] """ | if not numbers: return [] result = [] for n in numbers[:-1]: result.append(n) result.append(delimeter) result.append(numbers[-1]) return result | (Not pass, PPL:2.45) def intersperse (numbers: List[int], delimeter: int) -> List[int]: """ Insert a number 'delimeter' between every two consecutive elements of input list 'numbers' >>> intersperse([], 4) [] >>> intersperse([1, 2, 3], 4) [1, 4, 2, 4, 3] """ answer=[] for i in range(0,len(numbers),1): answer = answer + [ numbers[i] ] answer = answer + [ delimeter ] return answer | (Not pass, PPL:3.38) Maybe you meant () -> List[int] |
| def is_prime(n): """"Return true if a given number is prime, and false otherwise. """ | if n ¡ 2: return False for k in range(2, n - 1): if n % k == 0: return False return True | (Pass, PPL:1.94) "'python def is_prime(n): if n == 1: return False for i in range(2, n): if n % i == 0: return False return True | (Pass, PPL:1.78) def is_prime(n): if n == 1: return False for i in range(2, n): if n % i == 0: return False return True if __name__ == '__main__': import doctest doctest.testmod() |

et al., 2024), we set the lexical diversity to 60 without considering order diversity. Additionally, we increased the order diversity by 20 for DIPPER-2 following previous work (Liu et al., 2023a).

## G RELATED WORK

Existing white-box watermarking techniques fall into two categories: watermarking during logits and probabilities generation, and watermarking by controlling sampling strategies.

**Watermarking during logits and probabilities generation.** This category of watermarking methods inserts watermarks into LLMs by artificially adjusting the raw logits or probabilities generated by the LLM. Among this category, Kirchenbauer et al. (2023a) propose the first watermarking method based on logits adjustment. Their approach randomly partitions the vocabulary set into a green and a red list at each generation step, increasing the logits of green list tokens while keeping red list tokens' logits fixed. Lee et al. (2023) extend the green and red list-based watermarking method to low-entropy scenarios. They adjust the logits only during high-entropy generation steps, leaving the raw logits unchanged for low-entropy steps. Ren et al. (2023) improve the vocabulary set partition process by determining the green and red lists based on semantic embeddings of preceding tokens rather than their hash values. Fernandez et al. (2023) propose a multi-bit watermarking method that generates a multi-dimensional vector at each generation step, which is utilized to modify logits produced by the original LLM. Their approach allows embedding any bit of watermarking information, up to the dimension of the vector used in the logits adjustment. Yoo et al. (2023) develop a multi-bit method by extending the two-list partition idea to multi-list partitions. At each generation step, the vocabulary set is divided into multiple lists. Based on the message to be inserted, the logits for tokens in a selected list are increased, while the token logits in all other lists remain unchanged.

Instead of splitting the vocabulary set into different lists, Hu et al. (2024) introduce a method that randomly shuffles the order of all token probabilities within the interval $[0, 1]$, setting the probabilities in the first half of the interval to 0 and doubling those in the second half. During the detection phase,

a likelihood ratio test examines the significance of the likelihood that the given text is generated with the adjusted probability distribution. Wu et al. (2024) further generalizes this method by introducing a hyperparameter $\alpha \in [0, 0.5]$, which controls the two cutoff points $\alpha$ and $1 - \alpha$ within the interval $[0, 1]$. The probability masses for the three resulting sub-intervals are adjusted accordingly.

**Watermarking by controlling sampling strategies.** This category of watermarking methods inserts watermarks into the token sampling process by using watermark information to control the sampling of candidate tokens. For example, Christ et al. (2023) introduce a watermarking method that represents each token in the vocabulary set as a binary string of 0s and 1s. Next, a sequence of values from 0 to 1 is sampled uniformly. These values guide the token sampling process: if the predicted probability for a position in the binary string is larger than the corresponding pseudo-random value, that position is assigned a 1; otherwise, it is assigned a 0. Once all positions are determined, the token corresponding to the resulting binary string is sampled. Additionally, previous work (Kuditipudi et al., 2023) use a sequence of values randomly sampled from a uniform distribution between 0 and 1. The value controls the token sampling process through a decoder function, where the decoder function varies based on the sampling strategy. Hou et al. (2023) sample new sentences according to the original LLM until a sentence's semantic value falls into the acceptance region. The acceptance region is predefined by randomly splitting the space of semantic embedding according to the context and the key.

