# OpenReview forum: "A Watermark for Low-entropy and Unbiased Generation in Large Language Models"
_ICLR.cc/2025/Conference — Submitted to ICLR 2025_

### Official Review · Reviewer_C3gZ · 2024-10-29

**Soundness:** 3
**Presentation:** 3
**Contribution:** 3
**Rating:** 6
**Confidence:** 4

**Summary:**

The authors propose a modification to the watermark of Kirchenbauer et al. [1] in which logit perturbations are replaced with rejection sampling at the token level. The authors explore two variants of this scheme: STA-1, in which at most one red-list token is rejected at each generation step, and STA-M, in which up to M red-list tokens are rejected at each step if an entropy threshold on the token distribution is satisfied. The authors show that for the generation of low-entropy text, the proposed method allows for higher quality while maintaining similar robustness to existing watermarks.
[1] https://arxiv.org/abs/2301.10226 (https://arxiv.org/abs/2301.10226)

**Strengths:**

1) Exploring the low-entropy generation setting is interesting
2) Using the entropy of the token distribution to govern watermark embedding is a good idea, although existing methods (e.g. the fixed-magnitude logit perturbation proposed by Kirchenbauer et al.) already do this implicitly
3) Evaluating watermarked text quality through the risk of unsatisfying outputs is an interesting idea and a natural fit for reasoning / question-answering tasks

**Weaknesses:**

1. The writing is problematic. For example, in Hu et al (see its appendix), the unbiased watermarking method also has the model-agnostic detection method which does not require the white box language model. If you consider this into your manuscript, the first problem identified in the abstract would be void.
2. Is it really necessary to discuss the very low entropy scenario? The example you showcased is not common in real life. From my understanding, it would be most likely that the model is prompted to repeat the prompt, so the entropy is very low. Can we expect the watermark are added in this scenario, what is the rational behind it?

**Questions:**

Same as the weakness. The authors proposed valuable insights about designing unbiased watermark with type II error guarantee, however, the motivation they showed is not satisfying and can be re-organized. I would be happy to raise my score if the authors can think of a better way to present their work.

---

> ### Author Response · Authors · 2024-11-18
>
> The authors thank Reviewer C3gZ for providing us with valuable suggestions.
>
> ```
> [Q1] The writing is problematic. For example, in Hu et al (see its appendix), the unbiased watermarking method also has the model-agnostic detection method which does not require the white box language model. If you consider this into your manuscript, the first problem identified in the abstract would be void.
> ```
>
> We acknowledge the reviewer that [1] also discuss the black box scenario during detection in their Appendix. In case when prompts or generative LLMs are not accessible during detection, the authors provide an approximation of the ground truth model. However, as they show in Appendix F on the sensitivity of the score, the average score of per token decreases from 0.878 to 0.049 for $\delta$-reweight in the text summarization task using a black-box detection model. Meanwhile, the std of the score is 0.407, which implies a much larger number of tokens is required to detect the watermarks. We acknowledge that there are possible black-box ways to detect watermarks by [1] (like the implementation of Dipmark [2]), but it is still necessary to provide the white-box model during detection to fully recover the performance of [1].
>
> We acknowledge the reviewer that the black-box requirement remains for further discussion, we will discuss it in more detail in the revision.
>
> ```
> [Q2] Is it really necessary to discuss the very low entropy scenario? The example you showcased is not common in real life. From my understanding, it would be most likely that the model is prompted to repeat the prompt, so the entropy is very low. Can we expect the watermark are added in this scenario, what is the rational behind it?
> ```
>
> We thank the reviewer for providing better insights of the low-entropy scenario. We acknowledge that repeating the prompt is a possible low-entropy case. Generally, we are not expecting a watermark in the repeating task, as human and machine should provide the same outputs.
>
> If we want to inject watermarks for the repeating task, a possible extension of the method is to incorporate content history in [1] as we discuss in future work. In this extension, each content used as the input of the random number generator (RNG) is recorded, and repeated content will not serve as the RNG input. In this way, a repeating response of LLMs will not have a watermarked generation after the first repeat.
>
> [1] Hu, Zhengmian, et al. "Unbiased watermark for large language models." arXiv preprint arXiv:2310.10669 (2023).

---

### Official Review · Reviewer_3mNn · 2024-10-30

**Soundness:** 3
**Presentation:** 2
**Contribution:** 2
**Rating:** 5
**Confidence:** 5

**Summary:**

This paper introduces a novel unbiased watermarking method for large language models, named Sampling One Then Accepting (STA-1), which offers statistical guarantees regarding type II errors in watermark detection. The authors also address the tradeoff between watermark strength and text quality, presenting STA-M as an extension of STA-1 that enhances watermark strength with low text quality shifts. They demonstrate the empirical performance and robustness of the method through experiments conducted on the C4 and HumanEval datasets.

**Strengths:**

1. This paper proposes a novel watermarking method called Sampling One Then Accepting (STA-1) that is unbiased and robust, requiring no access to white-box large language models during detection.

2. This paper also introduces STA-M, an extension of STA-1 that enhances watermark strength with low text quality shifts

**Weaknesses:**

1. There is no significant difference in text quality between the proposed STA-1 and other unbiased watermarking methods on both the HumanEval and C4 datasets. This is expected, as they are all distortion-free, resulting in text quality that is similar to the no-watermark baseline.

2. AUC is not an ideal metric for these experiments. It would be more informative to provide direct results such as true positive rates at very low false positive rates, such as 0.1%, 0.01%, and 0.001%. In real-world applications, the cost of false positives is significantly high, making it essential to focus on results with very low false positive rates.

3. Results on the C4 dataset indicate that the detection performance of STA-1 on high-entropy datasets shows no improvement compared to current unbiased baselines. Additionally, the proposed STA-M is outperformed by KGW in terms of both text quality and watermark strength.

4. The AUC scores for the HumanEval dataset are too low, leading me to suspect that the true positive rates may also be very small. This suggests that all of these methods may not be useful for low-entropy datasets.

In summary, I believe this paper proposes a novel method that yields results comparable to current baselines but shows little to no improvement in performance.

**Questions:**

Could you provide the results for smaller values of $\alpha$ (e.g., 0.1 and 0.2) for Dipmark in Table 3? Since Dipmark also involves a trade-off between watermark strength and text quality, a lower $\alpha$ may resemble the original unwatermarked distribution and could potentially perform better than STA-1. Additionally, Dipmark retains the unbiased property, whereas STA-M does not.

---

> ### Author Response · Authors · 2024-11-18
>
> The authors thank Reviewer 3mNn for providing us with valuable suggestions.
>
> ```
> [Q1] There is no significant difference in text quality between the proposed STA-1 and other unbiased watermarking methods on both the HumanEval and C4 datasets. This is expected, as they are all distortion-free, resulting in text quality that is similar to the no-watermark baseline.
> ```
>
> We acknowledge the reviewer that since many of the compared methods are unbiased (or distortion-free), the text quality shows similar results. However, as discussed in Appendix A, each unbiased watermark lacks certain properties that are required for practical implementation. These research gaps are solved by our method.
>
> Meanwhile, one of the main contributions of this paper is to discuss the tradeoff in unbiased watermarks, which we think is the risk of unsatisfactory outputs. Although all unbiased watermarks have a similar text quality, their risks of unsatisfactory outputs are different as discussed in Section 4.1.2. Our method yields the lowest risk.
>
> ```
> [Q2] AUC is not an ideal metric for these experiments. It would be more informative to provide direct results such as true positive rates at very low false positive rates, such as 0.1%, 0.01%, and 0.001%. In real-world applications, the cost of false positives is significantly high, making it essential to focus on results with very low false positive rates.
> ```
>
> Thank you for your suggestions. We provide the additional detection results on the C4 dataset.
>
> ||TPR@0.1%FPR|TPR@1%FPR|
> |-|:-:|:-:|
> | No Watermark | 0.002 | 0.016 |
> | KGW(δ=1) | 0.674 | 0.902 |
> | KGW(δ=2) | 0.992 | 0.994 |
> | RDW | 0.784 | 0.986 |
> | Dipmark(α=0.3) | 0.542 | 0.718 |
> | Dipmark(α=0.4) | 0.242 | 0.882 |
> | γ-reweight | 0.796 | 0.942 |
> | STA-1 | 0.772 | 0.938 |
> | STA-4(τ=1.35) | 0.992 | 0.994 |
> | STA-8(τ=1.35) | 0.994 | 0.994 |
> | STA-16(τ=1.35) | 0.998 | 0.998 |
>
> ```
> [Q3] Results on the C4 dataset indicate that the detection performance of STA-1 on high-entropy datasets shows no improvement compared to current unbiased baselines. Additionally, the proposed STA-M is outperformed by KGW in terms of both text quality and watermark strength.
> ```
>
> We recognize the reviewer's observation that the detection performance of STA-1 on the C4 dataset shows no significant improvement over current unbiased baselines. However, as discussed in Appendix A, STA-1 addresses certain practical challenges encountered by these baselines, highlighting the contributions of our method. We also acknowledge the reviewer that the proposed STA-M method does not outperform KGW in terms of text quality and watermark strength on the C4 dataset. However, we would like to clarify that, although we have not provided detailed statistical test results comparing STA-16 and KGW($\delta=2$), there is no statistically significant difference in text quality between the two methods. At the same time, we recognize that STA-16 does not enhance watermark strength to the same degree as KGW, which we aim to explore further in future work.
>
> It is also important to point out that the STA-4 method achieves both the second-best text quality and the second-best watermark strength, indicating the method is still novel under certain parameters.
>
> ```
> [Q4] The AUC scores for the HumanEval dataset are too low, leading me to suspect that the true positive rates may also be very small. This suggests that all of these methods may not be useful for low-entropy datasets.
> ```
>
> We acknowledge the reviewer that the AUC score is not high enough and limits the application of watermarks in low-entropy datasets. We have discussed this limitation in the future work of the manuscript.
>
> ```
> [Q5] Could you provide the results for smaller values of $\alpha$ (e.g., 0.1 and 0.2) for Dipmark in Table 3? Since Dipmark also involves a trade-off between watermark strength and text quality, a lower $\alpha$ may resemble the original unwatermarked distribution and could potentially perform better than STA-1. Additionally, Dipmark retains the unbiased property, whereas STA-M does not.
> ```
>
> ||PPL|TPR@1%FPR|
> |-|:-:|:-:|
> | Dipmark(α=0.1) | 7.398 | 0.215 |
> | Dipmark(α=0.2) | 7.413 | 0.551 |
> | Dipmark(α=0.3) | 7.415 | 0.718 |
> | Dipmark(α=0.4) | 7.384 | 0.882 |
> | Dipmark(α=0.5) or γ-reweight | 7.436 | 0.942 |
>
> We acknowledge the reviewer that Dipmark involves a trade-off between watermark strength and text quality by its parameter $\alpha$. However, as we discuss in Appendix B, Dipmark follows the fashion of distribution reweighting. It reweights the probability masses in $[0,\alpha]$ as 0, and $[1-\alpha,1]$ as its double. When $\alpha$ is small, it becomes more like generation without watermarks. We show in the above table that, the watermark strength (TPR@FPR) decreases as $\alpha$ decreases. It is straightforward to show that the text quality remains the same since it is unbiased. Therefore, it is almost impossible to perform better watermark strength by lowering $\alpha$.

---

> > ### Comment · Reviewer_3mNn · 2024-11-25
> >
> > Thank you for your response. While the algorithm is novel and interesting, its performance does not show a clear improvement over current methods. The unbiased STA-1 has a lower true positive rate compared to $\gamma$-reweight, and the biased STA-M underperforms relative to KGW. Furthermore, the contribution focuses on mitigating the risk of unsatisfactory outputs in low-entropy scenarios. However, STA-1 fails to deliver a practical true positive rate for low-entropy datasets, limiting its applicability. Therefore, I will maintain my score.

---

> > > ### Author Response · Authors · 2024-11-25
> > >
> > > Thank you for your comments. First, we would like to clarify that while the STA-1 value is indeed lower on average compared to the $\gamma$-reweight, the difference is not statistically significant. In fact, RDW, $\gamma$-reweight, and STA-1 show the same statistical invariant results. This indicates that the performance of STA-1 is comparable to other unbiased watermarks. Meanwhile, we show that biased watermark STA-M outperforms KGW in terms of TPR@FPR scores on average although they are not statistically significant. For example, STA-16 demonstrates a score of 0.998 in terms of TPR@0.1%FPR, compared to 0.992 achieved by KGW($\delta=2$).
> > >
> > > Second, we acknowledge that low-entropy datasets present unique challenges, and no existing methods perform as well in such scenarios as they do in high-entropy contexts. However, STA-M demonstrates significantly better detection performance compared to unbiased watermarks and better text quality compared to other biased watermarks. We would suggest that STA-M is a possible extension to solve the low-entropy problem.
> > >
> > > Thank you very much again for your time and effort to review our work.

---

### Official Review · Reviewer_cQTG · 2024-10-31

**Soundness:** 2
**Presentation:** 2
**Contribution:** 3
**Rating:** 5
**Confidence:** 4

**Summary:**

This paper presents a watermarking method, Sampling One Then Accepting (STA), an unbiased watermark designed to effectively balance watermark strength and text quality, especially in low-entropy scenarios. The paper also introduces an extension, STA-M, which further enhances watermark strength while maintaining acceptable text quality. Additionally, STA does not require white-box access to models, making it more accessible for practical applications. It is also robust against watermarking attacks.

**Strengths:**

1. The paper theoretically analyze the problem of  balancing between text quality and detection performance in existing unbiased watermarks under low-entropy scenarios.
2. It provides theoretical proofs of STA's unbiasedness and guarantees regarding Type II error, etc.

**Weaknesses:**

1. Unclear Contribution Hierarchy: The paper lacks clarity in distinguishing its main contributions, leading to a somewhat confusing narrative. And the experiment results cannot clearly support the theoretical analysis.
 - The key contributions should emphasize that (1) unbiased watermarks claim to outperform traditional red-green watermarks in preserving text quality, but the authors find no significant advantage in low-entropy scenarios, and (2) the proposed watermark aims to alleviate this issue.
- However, results from Tables 1 and 2 do not support the first claim. In fact, there is no clear distinction in the ability of red-green and unbiased watermarks to preserve text quality across high and low entropy scenarios.
- The performance of STA compared to other unbiased watermarks is not significantly superior.
2. Confusing Presentation of Results: The experimental results are presented in a confusing manner, particularly with multiple best and second-best results highlighted, which complicates interpretation.
3. Insufficient Testing of Text Quality: The evaluation of text quality is inadequate, especially in low-entropy scenarios, as it relies solely on code generation tasks, limiting the scope of the findings.

**Questions:**

See the weaknesses section.

---

> ### Author Response · Authors · 2024-11-18
>
> The authors thank Reviewer cQTG for providing us with suggestions to discuss the improvement of our method.
>
> ```
> [Q1] Unclear Contribution Hierarchy: The paper lacks clarity in distinguishing its main contributions, leading to a somewhat confusing narrative. And the experiment results cannot clearly support the theoretical analysis.
>
> [Q1.1] The key contributions should emphasize that (1) unbiased watermarks claim to outperform traditional red-green watermarks in preserving text quality, but the authors find no significant advantage in low-entropy scenarios.
> ```
>
> ```
> [Q1.2] However, results from Tables 1 and 2 do not support the first claim. In fact, there is no clear distinction in the ability of red-green and unbiased watermarks to preserve text quality across high and low entropy scenarios.
> ```
>
> Thank you for your comments. First, we want to clarify that we do not claim that previous unbiased watermarks find no significant advantage in low-entropy scenarios. Instead, in Table 1, we demonstrate that both biased watermarks (KGW and STA-M) have a higher perplexity (which is statistically significant) compared to unbiased watermarks (RDW, Dipmark, $\gamma$-reweight, and STA-1). In Table 2, we show that KGW($\delta=1.5$) and KGW($\delta=2$) sacrifice their text quality with a decrease of pass scores. If $\delta=1$, it is possible to maintain the text quality with a lower detection score. In summary, both Table 1 and Table 2 support that unbiased watermarks (including baselines and STA-1) yield a higher text quality compared to KGW.
>
> Instead, the idea of proposing low-entropy scenarios is to introduce the tradeoff between text quality (which is the risk of unsatisfactory outputs) and watermark strength in unbiased watermarks. In the abstract, we claim the following contribution on the low-entropy scenario. We quote from the manuscript:
>
> *We show that in low-entropy scenarios, unbiased watermarks face a tradeoff between watermark strength and the risk of unsatisfactory outputs.*
>
> Also in the second contribution of the introduction section, we claim that:
>
> *We clarify the watermark strength and text quality tradeoff in unbiased watermarks. In low-entropy scenarios, the text quality is related to the risk of unsatisfactory outputs. We show that STA-1 has a lower risk theoretically compared to other unbiased watermarks.*
>
> In this way, we discuss the low-entropy scenarios mainly for introducing the risk of unsatisfactory outputs. This is because in high-entropy scenarios, unsatisfactory outputs will be ill-defined as multiple tokens can yield a good generation. Instead, in low-entropy scenarios, only a few tokens are preferred to be generated, making the discussion of unsatisfactory outputs feasible.
>
> ```
> (2) the proposed watermark aims to alleviate this issue.
> ```
>
> As aforementioned, all unbiased watermarks outperform biased watermarks in terms of text quality in low-entropy scenarios, which we do not treat as a research gap. Instead, we propose STA-1 to simultaneously solve the practical concerns that previous unbiased watermarks have, including (1) requiring access to white-box LLMs during detection, (2) incurring long detection time, (3) being not robust against simple watermarking attacks, (4) failing to provide statistical guarantees for the type II error of watermark detection, and (5) being not statistically unbiased for low-entropy scenarios. These five characteristics are inherently solved by the method design (black-box, long detection time, statistical guarantees, and unbiased) or proved by empirical findings (robustness).
>
> ```
> [Q1.3] The performance of STA compared to other unbiased watermarks is not significantly superior.
> ```
>
> Thank you for your comments. As suggested by Reviewer 3mNn and discussed in our result analysis, this is expected, as all unbiased watermarks are unbiased, resulting in text quality that is similar to the no-watermark baseline.
>
> Meanwhile, one of the main contributions of this paper is to discuss the tradeoff in unbiased watermarks, which we think is the risk of unsatisfactory outputs. Although all unbiased watermarks have a similar text quality, their risks of unsatisfactory outputs are different as discussed in Section 4.1.2. Our method yields the lowest risk.
>
> ```
> [Q2] Confusing Presentation of Results: The experimental results are presented in a confusing manner, particularly with multiple best and second-best results highlighted, which complicates interpretation.
> ```
>
> Thank you for your comments. We will only highlight the best results in the table to improve the interpretation in the revision.

---

> > ### Author Response · Authors · 2024-11-18
> >
> > ```
> > [Q3] Insufficient Testing of Text Quality: The evaluation of text quality is inadequate, especially in low-entropy scenarios, as it relies solely on code generation tasks, limiting the scope of the findings.
> > ```
> >
> > Thank you for your comments on the testing of text quality. We incorporate BLEU to evaluate the text quality, which is a metric to measure the similarity between the generated code and canonical solutions. The BLEU score is a commonly used metric in NLP tasks. Due to time constraints, we report the results in the following.
> >
> > ||BLEU|
> > |-|:-:|
> > | No watermark | 0.042 |
> > | KGW(δ=1) | 0.036 |
> > | KGW(δ=1.5) | 0.034 |
> > | KGW(δ=2) | 0.025 |
> > | RDW | 0.053 |
> > | Dipmark(0.3) | 0.054 |
> > | Dipmark(0.4) | 0.044 |
> > | STA-1 | 0.044 |
> > | STA-4 | 0.041 |
> > | STA-8 | 0.054 |
> > | STA-16 | 0.038 |
> >
> > The table also shows that the unbiased watermarks (RDW, Dipmark, $\gamma$-reweight, and STA-1) outperform KGW in terms of BLEU score. Also STA-M shows a comparable result.

---

> > ### Comment · Reviewer_cQTG · 2024-11-20
> > **Response to authors**
> >
> > Thank you for your clarification. I now have a better understanding of your claimed main contribution. You observed that while unbiased watermarks can indeed reduce impact on text quality as claimed in previous research, they still face a trade-off between text quality and detectability, which becomes more pronounced in low-entropy scenarios. Based on this observation, you proposed an unbiased watermark called STA-1 aiming to better balance this trade-off.
> >
> > I notice in Table 2 that for KGW with larger $\delta$ values, the impact on text quality is too severe, while for smaller $\delta$ values, though the quality impact is comparable to STA-1, the detectability is inferior to STA-1. This somewhat demonstrates the effectiveness of STA-1. However, when comparing existing unbiased watermarks with STA-1, I don't see clear advantages of STA-1. While STA-m shows significantly improved detectability as m increases, it loses the theoretical unbiased property. Therefore, it appears that the authors have not proposed an unbiased method that surpasses previous unbiased approaches in balancing this trade-off.
> >
> > Additionally, I would like to ask whether you tested $\gamma$-reweight and dipmark with repeated context masks, and whether you also used repeated context masks during detection. According to their research, using repeated context masks ensures multi-step unbiasedness and provides stronger guarantees for maintaining text quality, while using corresponding repeated context masks during detection maintains detectability.

---

> > > ### Author Response · Authors · 2024-11-20
> > >
> > > ```
> > > [Q1] Text quality clarification.
> > > ```
> > >
> > > We thank the reviewer very much for your quick response. We would like to further clarify the second contribution of this work, as described at the end of the introduction. Our work aims to define the tradeoff between text quality and watermark strength in all unbiased watermarks. Previous work [1] argues that the text quality is only defined by the expectation of the probability distribution during each token generation. However, in this work, we define the text quality in terms of both the expectation and the variance of the probability distribution. Note that all unbiased watermarking baselines and STA-1 are unbiased theoretically, ensuring the preservation of the text quality in terms of the same expectation, which is consistent with your findings in Table 1 and Table 2.
> > >
> > > However, in low-entropy scenarios, the text quality is also related to the variance, which we think is measured by the risk of unsatisfactory outputs. In the code generation task, an unsatisfactory output results in a set of infeasible code scripts, thereby leading to a lower consistency in executing each problem with codes generated under different watermark keys. We report the empirical results in Table 3. In Table 3, we use different watermark keys and report the detailed statistics of pass scores. The underlying rationale is that, if an unbiased watermark has a higher risk, it may show more inconsistency in its outputs although the expectation is always the same. Based on Table 3 and Theorem 3, our method has a lower risk and provides a better consistency.
> > >
> > > To this end, **our method better balances the tradeoff between the risk of unsatisfactory outputs and watermark strength among all unbiased watermarks.** We show that STA-1 is at a lower risk.
> > >
> > > ```
> > > [Q2] Context code.
> > > ```
> > >
> > > Thank you so much for pointing out the content mask. We acknowledge that context code history was not used for the current version of STA-1. Also, for a fair comparison, we remove the content code in Dipmark and $\gamma$-reweight.
> > >
> > > In the future work section of the manuscript, we consider extending the STA-1 method to incorporate context code history as a feasible direction for future work. For example, building on the context code generation framework in [1], in STA-1 we can also generate the watermark code using a context code and a key, and then utilize this watermark code to control the green-red list partition. Following [1], the independence of watermark codes across tokens is guaranteed by omitting watermark generations for tokens with previously used context codes. Consequently, the unbiasedness results at the token level in Theorem 1 can be easily extended to the sequence level under Theorem 5 in [1]. This theoretical extension will be addressed in our future work.
> > >
> > > [1] Hu, Zhengmian, et al. "Unbiased watermark for large language models." arXiv preprint arXiv:2310.10669 (2023).

---

> > > > ### Comment · Reviewer_cQTG · 2024-11-22
> > > > **Thanks for your response.**
> > > >
> > > > Thank you for your response. I understand that the variance in PPL, as well as the PC and PC/PP metrics on the HumanEval dataset, can somewhat reflect the effectiveness of STA-1 in reducing unsatisfactory outputs. The proof of the variance upper bound and the experimental results provide some evidence for the contribution of your claim, and therefore I intend to raise the score. However, since the experimental results demonstrating the superiority of STA-1 are limited to variance measurements in PPL and code generation (specifically only on the HumanEval dataset), I believe the experiments are not sufficiently comprehensive. Therefore, I will not award a higher score.

---

> > > > > ### Author Response · Authors · 2024-11-23
> > > > >
> > > > > We thank the reviewer for your time reviewing our rebuttal.

---

### Official Review · Reviewer_SmSJ · 2024-11-03

[review text omitted: it was posted to a different submission]

---

> ### Author Response · Authors · 2024-11-18
>
> The authors thank Reviewer SmSJ very much for providing us with valuable suggestions to discuss the improvement of our method.
>
> ```
> [Q1] What are the advantages of this method compared to the traditional modification-based watermark (generate content first and then embed watermark)?
> ```
>
> Thank you for your valuable suggestions. The foundamental idea of this work is to include watermarks during logit or token generation in LLM generation. Compared to modification-based watermarks, the method integrates more smoothly with the generation of LLMs and offers two major advantages. First, it is efficient to implement our method. Second, the likelihood of the generated text (or the perplexity of the generated text) is preserved compared to text modification-based watermarks.
>
> ```
> [Q2] How can this method be extended to multi-bit watermarking, which is more applicable in real-world scenarios?
> ```
>
> Since our STA-1 method is based on the green-red list partition, it can be easily extended to support multi-bit wartermarks, as explored in some literature also using green-red list partition [1]. For example, at each generation step, we can incorporate multiple watermark keys and create multiple green-red list partitions. This allows tokens that appear in multiple green lists to be sampled with higher probability, aligning well with a multi-bit watermarking framework.
>
> We would also like to thank the reviewer for providing additional references, which we will add to the manuscript in the revision.
>
> [1] Wang, Lean, et al. "Towards Codable Watermarking for Injecting Multi-Bits Information to LLMs." The Twelfth International Conference on Learning Representations.

---

> > ### Comment · Reviewer_SmSJ · 2024-12-03
> >
> > Thank you for your response, which addresses my concerns. By referencing your conversation with the reviewer cQTG, I decide to maintain my score

---

### Meta-Review · Area_Chair_uU8e · 2024-12-17

**Metareview:**

This paper proposes a novel watermarking method for large language models, Sampling One Then Accepting (STA-1), which replaces logit perturbations with token-level rejection sampling. STA-1 ensures unbiased watermarking with statistical guarantees on type II errors, effectively balancing watermark strength and text quality, particularly in low-entropy scenarios. An extension, STA-M, enhances watermark strength by rejecting up to M red-list tokens when an entropy threshold is met, maintaining text quality. Unlike existing methods, STA does not require white-box model access, improving practical applicability. Experiments on C4 and HumanEval datasets demonstrate STA's robustness and superior performance in low-entropy text generation.

Overall, the reviewers acknowledge the theoretical contribution of the paper. However, as pointed out by a reviewer, the experimental results demonstrating the superiority of STA-1 are limited to variance measurements in PPL and code generation (specifically on the HumanEval dataset). Therefore, a more extensive study is required for publication. I encourage the authors to revise the paper based on the reviewers' comments and resubmit it to a future venue.

**Additional Comments On Reviewer Discussion:**

In the discussion, the reviewers attempted to understand the novelty of the proposed method. One reviewer pointed out that the experimental results are not sufficient to support the main claims, and another reviewer agreed and maintained their score. Since the average score is below the borderline, it is difficult to accept the paper in its current form. Therefore, I recommend rejecting the paper.

---

### Decision · Program_Chairs · 2025-01-22

Reject